# Discovering Clone Negatives via Adaptive Contrastive Learning for Image-Text Matching

**Renjie Pan, Jihao Dong, Hua Yang**[*]
Institute of Image Communication and Network Engineering, Shanghai Jiao Tong University
Shanghai Key Lab of Digital Media Processing and Transmission, Shanghai Jiao Tong University
`{rjpan21,dongjihao,hyang}@sjtu.edu.cn`

## Abstract

In this paper, we identify a common yet challenging issue in image-text matching, i.e., clone negatives: negative image-text pairs that semantically resemble positive pairs, leading to ambiguous and sub-optimal matching outcomes. To tackle this issue, we propose Adaptive Contrastive Learning (AdaCL), which introduces two margin parameters along with a modulating anchor to dynamically strengthen the compactness between positives and mitigate the influence of clone negatives. The modulating anchor is selected based on the distribution of negative samples without the need for explicit training, allowing for progressive tuning and advanced in-batch supervision. Extensive experiments across several tasks demonstrate the effectiveness of AdaCL in image-text matching. Furthermore, we extend AdaCL to weakly-supervised image-text matching by replacing human-annotated descriptions with automatically generated captions, thereby increasing the number of potential clone negatives. AdaCL maintains robustness in this setting, alleviating the reliance on crowd-sourced annotations and laying a foundation for scalable vision-language contrastive learning.

## 1 Introduction

Image-text matching(Fang et al., 2023; Qu et al., 2021; Huang et al., 2019; Shi et al., 2019; Chen et al., 2021) has become de facto the most fundamental multimodal task in recent years, which involves searching for images with text queries (text-image) and the retrieval of sentences using images (image-text). Existing image-text matching methods usually learn a robust cross-modal representation through various fusion paradigms, such as object-level or global-level aligning(Zhang et al., 2022; Wang et al., 2022a; Cheng et al., 2022; Li et al., 2022b; Qu et al., 2021; Chen et al., 2021; Wang et al., 2020a; Wu et al., 2019; Li et al., 2019; Lee et al., 2018; Dong et al., 2024; Song et al., 2024). On the other hand, contrastive learning(Li et al., 2022a; Radford et al., 2021; He et al., 2020; Oord et al., 2018; Zhang & Lu, 2018) has been widely adopted as the objective, which brings similar (or positive) samples closer in the latent space while pushing dissimilar (or negative) samples apart. Among them, Triplet ranking loss (TRL)(Wang et al., 2019; 2018; Faghri et al., 2017) and Contrastive loss (CL)(Li et al., 2021; Radford et al., 2021; He et al., 2020) have witnessed the success of vision-language contrastive learning. Specifically, TRL preserves the relative distance between in-batch samples through a manually-set fixed margin. CL is defined as the cross-entropy of the softmax-normalized similarity between each image-text pair. In self-supervised learning and multimodal learning, InfoNCE and its variants(Oord et al., 2018; Ma & Collins, 2018; Liang et al., 2024; Cheng et al., 2020; Zhang & Lu, 2018) are the prevalent forms of CL.

Despite these advancements, current contrastive learning for image-text matching faces an inherent challenge: During an in-batch learning, it is common for image-text pairs to possess semantically related text annotations and similar visual cues, thereby rendering them practically indistinguishable (reflected by their closing similarity scores). As shown in Figure 1(a), $T_2$ is a typical hard negative for $I_1$, which has been widely discussed (Pan et al., 2023a; Wang et al., 2020b; Zhang et al., 2022). It is noteworthy that apart from the key semantic "two skyscrapers", $T_3$ is semantically-consistent with $I_1$. Although matching $I_1$ and $T_3$ is indeed correct, it is no doubt a ***sub-optimal*** result because

---

[*]Corresponding author.

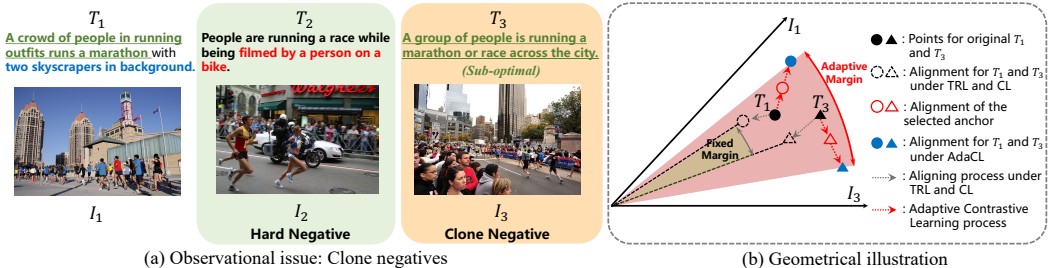

(a) Observational issue: Clone negatives          (b) Geometrical illustration

Figure 1: (a) A case of "hard negatives" and "clone negatives". Compared with gold label ($T_1$), clone negative ($T_3$) is defined as a "sub-optimal" text sample. (b) Geometrical illustration of Adaptive Contrastive Learning, which adaptively enlarges the distance between positives and clone negatives.

the ground-truth annotation $T_1$ exhibits more precise and fine-grained semantics, demonstrating a closer correspondence with $I_1$. Different from false negatives which should be considered as "true", we refer to samples like $T_3$ as **clone negatives**, which is quite challenging for matching in such scenarios. Distinguishing clone negatives through cross-modal fusion is highly intractable since clone negatives are somehow "positive". Also, the limited prior works primarily focus on negative mining strategies for single modality (Wang et al., 2024; Robinson et al., 2020; Yang et al., 2021), or learning noisy correspondence (Han et al., 2024; Huang et al., 2021). Image-text matching continues to predominantly rely on vanilla TRL or CL by setting a fixed margin during training, making it tough to actively address the issue of clone negatives since it has already pre-defined the label of all the image-text pairs (positive or negative).

In this work, we propose Adaptive Contrastive Learning (AdaCL) to exploit reliable clone negatives and distinguish them effectively in image-text matching. Instead of the fixed learning pattern of TRL and CL, we specifically add two margin parameters, serving as scaling and shifting factors, to synchronously enhance the compactness of positives while introducing the supervision of the clone negatives. To progressively tune the margin parameters, we propose to select a modulating *anchor* from each in-batch similarity score based on Gaussian discriminant analysis without explicit training. In this way, *anchor* effectively reflects the intensity of in-batch clone negatives, and imposes a substantial penalty through the margin parameters, thereby adaptively enlarging the distance between positives and clone negatives. Comprehensive experiments conducted on image-text matching, noisy correspondence learning, CLIP pre-training, and text-based person search demonstrate the effectiveness and robustness of AdaCL.

Moreover, AdaCL can be applied to weakly-supervised image-text matching with *structured and standardized text annotations*. We propose to increase the number of potential clone negatives by substituting the original textual descriptions with automatically generated captions using image captioning methods. We verify AdaCL by four distinct captioning tools, and it is observed that AdaCL can also learn robust cross-modal semantics. This not only demonstrates the effectiveness of AdaCL in distinguishing clone negatives, but also mitigating the need for crowd-sourced annotations, laying a foundation for future vision-language contrastive learning. To sum up, the main contributions of this paper include:

- We elucidate the issue of **clone negatives**: semantic-consistent but suboptimal matching candidates, and highlight the challenge they pose to existing contrastive learning for image-text matching.

- We propose Adaptive Contrastive Learning (AdaCL), introducing two margin parameters that dynamically propagates the semantics of clone negatives during training, enhancing the compactness of positives and mitigating the impact of clone negatives.

- AdaCL demonstrates superiority in both various downstream tasks and our proposed weakly-supervised image-text matching, which uses automatically generated captions as annotations. Extensive experiments prove the robustness of AdaCL, alleviating the reliance on manual annotations.

## 2 RELATED WORK

**Image-Text Matching.** The general process of Image-text matching can be attributed to two key components: (1) cross-modal feature fusion, and (2) learning objectives. The former can be categorized to global representation fusion and region-level fragment fusion. Global representation fusion extracts holistic feature from two modalities, and then mapping them to a joint visual-textual alignment where the similarity is calculated(Zhang & Lu, 2018; Ma et al., 2015). Region-level fragment fusion captures local content of two modalities to enhance the cross-modal interaction(Li et al., 2017a; Lee et al., 2018; Cheng et al., 2022; Diao et al., 2021; Pan et al., 2021). Apart from this, graph-structured methods (Liu et al., 2020; Wang et al., 2020a; Li et al., 2022b) also leverage nodes to represent region-level matching, further exploiting the instance relationships. The learning objectives of image-text matching can be categorized to contrastive loss (CL) (Li et al., 2022a; Radford et al., 2021; He et al., 2020; Oord et al., 2018; Zhang & Lu, 2018) and triplet ranking loss (TRL) (Wang et al., 2019; 2018). For CL, Zhang & Lu (2018); Wang et al. (2020b) proposed to minimize the distance between the projection distributions and the normalized matching distributions together with a classification objective. TRL is widely adpoted by existing methods (Wang et al., 2018; Pan et al., 2023a; Qu et al., 2021). Several studies have also sought to address false negatives by reducing sampling weights(Li et al., 2023a; Zhang et al., 2022; Li et al., 2023c), which is essentially a "passive" approach. However, such methods cannot handle clone negatives due to the fundamental differences between false negatives and clone negatives, as the latter are of significant learning value. Also, both TRL and CL are inherently limited by a manully-set margin, which fails to consider the sample discrepancy of clone negatives. To address this challenge, a pivotal solution lies in the ability to adaptively emphasize the learning of these samples.

**Contrastive Learning.** Contrastive learning has emerged as a powerful paradigm in various fields, such as self-supervised learning(He et al., 2020; Grill et al., 2020), representation learning(Radford et al., 2021), and recommendation systems(Shan et al., 2024; Lin et al., 2024; 2025; Shan et al., 2025). It aims to learn discriminative features by contrasting positive pairs against negative pairs (Schroff et al., 2015; Oord et al., 2018; Zhang & Lu, 2018). The foundation of contrastive learning can be traced back to metric learning, where the goal was to learn a distance function that better represents the underlying structure of the data (Bromley et al., 1993; Taigman et al., 2014; Wang et al., 2017; Grill et al., 2020; Caron et al., 2020). Seminal works such as SimCLR (Chen et al., 2020b) and MoCo (He et al., 2020) established the foundation for contrastive learning in computer vision. The former introduced a simple yet effective approach using data augmentation and a large batch size, while the latter proposed a momentum encoder to maintain a consistent dictionary of negative samples. In parallel to its success in computer vision, contrastive learning has also shown promise in multimodal learning (Radford et al., 2021; Li et al., 2022a; 2021), which employed a contrastive objective to align text and image representations in a shared latent space. By pretraining on large-scale text-image pairs, remarkable zero-shot transfer capabilities can be achieved. Despite its success, the requirement for high-quality datasets is still a significant hurdle for many real-world applications(Zhang et al., 2024b; Young et al., 2014) filling with clone negatives, which hinder a broader adoption of contrastive learning.

## 3 ADAPTIVE CONTRASTIVE LEARNING

Our aim is to synchronously enhance the compactness between positive samples and calibrating the supervision of clone negatives. We begin by revisiting vanilla contrastive learning, followed by our proposed Adaptive Contrastive Learning (AdaCL) and the anchor selection methdology. By replacing TRL and CL with AdaCL, both the matching performance and robustness have been greatly improved, demonstrated by supervised and weakly-supervised image-text matching on various baselines.

### 3.1 PRELIMINARIES

**Architecture of contrastive learning**. The prerequisites for vision-language contrastive learning include textual and visual modalities. Given a text description with $a$ words and the pairwise image, the textual and visual backbone are denoted as $\mathcal{F}_t(\cdot)$ and $\mathcal{F}_v(\cdot)$. Each word vector is mapped to the common space through a fully connected layer. $\mathcal{F}_v$ generates a sequence of $n$ patches to extract visual feature. The fine-grained representation of both modalities can be denoted as: $\mathbf{w} = \mathcal{F}_t(T) \in \mathbb{R}^{a \times D_t}$ and $\mathbf{v} = \mathcal{F}_v(I) \in \mathbb{R}^{n \times D_v}$. The similarity score of a certain image-text pair can be denoted as

Figure 2: Illustration of Adaptive Contrastive Learning (AdaCL). Salient negatives and Reference clone negatives are selected by Salient Score (Sec. 3.3). AdaCL dynamically adjusts two introduced margin parameters based on GDA, progressively regulating the distance between positives and clone negatives (Sec. 3.2). Unlike vanilla CL, AdaCL enhances robustness by adapting parameters to the intensity of clone negatives. Additionally, AdaCL features a plug-and-play design, demonstrating excellent adaptability and compatibility across various tasks (Sec. 4).

$s(I, T) = g_v(\mathbf{v})^\mathsf{T} g_w(\mathbf{w})$. For clarity and readability, we use $g_v(\cdot)$ and $g_w(\cdot)$ to represent optional cross-modal operations (such as fusion or connector) between visual and textual representation, where $D$ is the dimension of the joint embedding space.

**Training objective.** Vanilla contrastive learning relies on a fixed learning pattern to narrow the distance between positive samples. Specifically, the softmax normalized similarity for image $I$ and its corresponding text $T_i$ can be denoted as:

$$p_i(I) = \frac{\exp(s(I, T_i)/\tau)}{\sum_{j=1}^{M+1} \exp(s(I, T_j)/\tau)}, \tag{1}$$

where the denominator consists of $M + 1$ samples, i.e., one positive and $M$ negatives. Noted that if momentum memory bank is not used, then $M$ equals to the mini-batch size $N$. $\tau$ is a fixed temperature parameter controlling the overall supervision. $p_i(I)$ represents the probability of assigning $I$ to the corresponding ground truth $T_i$. Then, a cross-entropy loss is adopted:

$$\mathcal{L} = \mathbb{E}_{I \sim D} \left[ \mathbb{H}(\mathbf{y}(I), \mathbf{p}(I)) \right] = -\frac{1}{N} \sum_{i=1}^{N} y_i(I) \log(p_i(I)), \tag{2}$$

where $\mathbf{y}(I)$ is a one-hot distribution where the ground-truth is set by 1. Equation 2 pulls positive image-text pairs closer while pushing away the *pre-defined* negative pairs.

## 3.2 ADAPTIVE CONTRASTIVE LEARNING

Current image-text matching typically utilize TRL(Wang et al., 2019; 2018) and CL(Zhang & Lu, 2018; Wang et al., 2020b; Radford et al., 2021) during training. As mentioned before, it is highly intractable to distinguish clone negatives, leading to insufficient representation learning. Also, the semantic discrepancy between positives and clone negatives cannot be fully exploited. Based on this fact, AdaCL is proposed by introducing two margin parameters that play the role of scaling and shifting factors. The margin parameters are adaptively tuned under the supervision of potential clone negatives selected via the distribution of in-batch similarity logits.

For simplicity, we analyze the $i$-th image-text pair from a mini-batch. In order to uncover and enhance the supervision of potential clone negatives, we introduce two margin parameters to $p_i(I)$ to serve as the scaling and shifting factors:

$$\hat{p}_i(I) = \frac{\exp \left[ m_1(s(I, T_i) - m_2) \right]}{\exp \left[ m_1(s(I, T_i) - m_2) \right] + \sum_{j=1, j \neq i}^{M+1} \exp \left[ s(I, T_j) \right]}, \tag{3}$$

$m_1$ and $m_2$ are the tuning targets during training. For the tuning process, we strategically select a specific *anchor* that embodies the characteristics of clone negatives. *anchor* serves as a regulation in the overall supervision of in-batch clone negatives (details in Section 3.3). In order to compute $m_1$ and $m_2$ during each batch learning, two conditions are specifically analyzed. First, the probability of *anchor* can be obtained using Equation 3:

$$\hat{p}_u = \frac{\exp\left[m_1(anchor - m_2)\right]}{\exp\left[m_1(anchor - m_2)\right] + \sum_{anchor}}, \tag{4}$$

where $\sum_{anchor}$ is the simplification of $\sum_{k=1,k\neq u}^{M+1} \exp\left[s(I_u, T_k)\right]$. Since *anchor* represents the ensemble average probability of potential in-batch clone negatives, its corresponding $\hat{p}_u$ reflects the approximate convergence degree of the model to a large extent. Second, if we can control $\hat{p}_u$, then the overall supervision within the batch are well regulated. Therefore, we propose to progressively tune $m_1$ and $m_2$ based on $\hat{p}_u$. To achieve this, a specific boundary condition of Equation 4 is analyzed: Since cross-entropy loss is represented as $L = -log(\hat{p}_u)$, when the score of *anchor*, i.e., $s(I_u, T_u)$ (same as $s^{pos}$ in Equation 11) approaches 1 (indicating a high similarity), a relative small $m_1$ would unnecessarily penalize the loss function for correct image-text pairs. Based on this fact, we reason out a condition that $\hat{p}_u$ should be as close to 1 as possible when $s(I_u, T_u)$ approaches 1, to boost distinguishing clone negatives. This condition can be expressed as:

$$\lim_{s(I_u, T_u)\to 1} \hat{p}_u \approx 1. \tag{5}$$

Combining Equation 4, we can approximate the following boundary:

$$\frac{\exp\left[m_1(1 - m_2)\right]}{\exp\left[m_1(1 - m_2)\right] + \sum_{anchor}} = 1 - \epsilon, \tag{6}$$

where $\epsilon$ is a small value added to avoid $m_1$ reaches 0. With Equation 4 and Equation 6, we can derive the values of $m_1$ and $m_2$ in Equation 3 iteratively and update $\mathcal{L}_{ada}$ based on the objective format in Equation 2 during training. In this way, $\mathcal{L}_{ada}$ effectively propagates the semantics of clone negatives via $m_1$ and $m_2$. The detailed derivation is in A.3 of the appendix.

## 3.3 ANCHOR SELECTION

As mentioned above, our aim is to select a salient *anchor* from the in-batch similarity scores to control the supervision of potential clone negatives. It is intuitive that the image/text with the highest similarity score with its positive text/image is more likely to be an salient negative, while a relatively lower score corresponds to a potential negative which is hard to distinguish. To best quantitatively analyze the in-batch similarity, we introduce a Salient Score for anchor selection. Specifically, $S_i = \{s_{ij}\}_{j=1}^{M+1}$ refers to the similarity between each in-batch text/image and the $i^{th}$ image/text, and $s_{ii}$ is the positive. Then, the salient score of each in-batch sample is defined as the difference between the positive and the average similarity of the negatives. Mathematically, it can be expressed as:

$$\text{Salient Score}_i = s_{ii} - \frac{1}{M}\sum_{j=1,j\neq i}^{M+1} s_{ij}. \tag{7}$$

Compared with directly choosing the sample with the highest similarity, salient score better reflects the "salient" extent by considering the negative similarity. The image-text pair with the highest salient score is termed as $S_{sln}$, including 1 positive and $M$ salient negatives. On the contrary, the image-text pair with the lowest salient score is termed as $S_{cln}$, including 1 positive and $M$ potential clone negatives.

To predict the potential clone negatives from each mini-batch, a straightforward and simple way is to calculate the salient score of the rest in-batch image-text pairs. However, we found that in contrastive learning, similarity scores typically do not exhibit significant numerical differences, thus applying salient score to all the samples may result in over-fitting. Therefore, we revisit and propose to leverage a classical algorithm Gaussian Discriminant Analysis (GDA) for prediction **without** the need for explicit training. Specifically, the classification probability can be formulated as follows:

$$p\left(\mathcal{C} \mid s\right) = \frac{p\left(s \mid \mathcal{C}\right) p\left(\mathcal{C}\right)}{p\left(s \mid \mathcal{C}\right) p\left(\mathcal{C}\right) + p\left(s \mid \bar{\mathcal{C}}\right) p\left(\bar{\mathcal{C}}\right)} = \frac{\exp\left(a_c\right)}{\exp\left(a_c\right) + \exp\left(a_{\bar{c}}\right)}. \tag{8}$$

Here a binary classification is adopted, thus $\mathcal{C}$ and $\bar{\mathcal{C}}$ represent the pairwise similarity is / is not a clone negative. $s$ is the similarity score. The logit function is $a_c = \log(p(s \mid \mathcal{C})p(\mathcal{C}))$. Therefore, the classifier can be obtained by analyzing the data distribution of clone negatives and its prior

distribution. In GDA, the features are typically assumed to follow the gaussian distribution with identical covariance. Since the dimension of similarity score is 1, i.e., $s \in \mathbb{R}$, the aforementioned distribution expression can be simplified to a univariate distribution, where $(s \mid \bar{\mathcal{C}}) \sim \mathcal{N}\left(\mu_{\bar{c}}, \sigma_{\bar{c}}^2\right)$, and $(s \mid \mathcal{C}) \sim \mathcal{N}\left(\mu_c, \sigma_c^2\right)$ Combining this assumption to Equation 8, we can obtain:

$$p\left(\mathcal{C} \mid s\right) = \frac{1}{1 + \frac{\pi_{\bar{c}}}{\pi_c} \frac{\sigma_c}{\sigma_{\bar{c}}} \exp\left[\frac{(s-\mu_c)^2}{2\sigma_c^2} - \frac{(s-\mu_{\bar{c}})^2}{2\sigma_{\bar{c}}^2}\right]}, \tag{9}$$

where $\pi_c$ and $\pi_{\bar{c}}$ are the priors of $\mathcal{C}$ and $\bar{\mathcal{C}}$ respectively. Given that $S_{sln}$ and $S_{cln}$ comprise the most representative salient negatives and potential clone negatives within a mini-batch, we utilize the negatives from $S_{sln}$ and $S_{cln}$ within each mini-batch to represent the empirical means and variances $\mu_c, \mu_{\bar{c}}, \sigma_c$, and $\sigma_{\bar{c}}$. The potential in-batch clone negatives are selected based on the criterion:

$$S^* := \{s \mid p\left(\mathcal{C} \mid s\right) > p\left(\bar{\mathcal{C}} \mid s\right)\}, \tag{10}$$

where $S^*$ is the estimated clone negative set from a mini-batch. Finally, *anchor* is obtained by the median of $\Delta S$, which is defined as:

$$\Delta S = \{\delta \mid \delta = |s^{pos} - s|\}, \forall s \in S^*; \ anchor := s^{pos} \mid \delta = \text{median}(\Delta S). \tag{11}$$

*anchor* reflects the average discrepancy between all potential clone negatives and their corresponding positive. With *anchor* and adaptive contrastive learning in Section 3.2, the overall process of AdaCL is demonstrated in Algorithm 1.

---

**Algorithm 1** Adaptive Contrastive Learning

---

**Input:** a mini-batch of $N$ image-text pairs, with $N$ positives and $N \cdot M$ negatives.
**Output:** $\mathcal{L}_{ada}$
 1: **for** each mini-batch **do**
 2:     Select in-batch grounding salient negatives and clone negatives;
      $S_{sln} = \{s_{i+j}\} \mid j \in M, i^+ = \text{argmax}_i \text{ Salient Score}_i$,
      $S_{cln} = \{s_{i-j}\} \mid j \in M, i^- = \text{argmin}_i \text{ Salient Score}_i$,
 3:     Sort out in-batch potential clone negatives $S^*$, and select *anchor* based on $\Delta S$;
      $p\left(\mathcal{C} \mid s\right) = 1/\left\{1 + \frac{\pi_{\bar{c}}}{\pi_c} \frac{\sigma_c}{\sigma_{\bar{c}}} \exp\left[\frac{(s-\mu_c)^2}{2\sigma_c^2} - \frac{(s-\mu_{\bar{c}})^2}{2\sigma_{\bar{c}}^2}\right]\right\}$,
      $S^* := \{s \mid p\left(\mathcal{C} \mid s\right) > p\left(\bar{\mathcal{C}} \mid s\right)\}$, *anchor* $:= s^{pos} \mid \delta = \text{median}(\Delta S)$,
 4:     Obtain the probability of *anchor* for tuning;
      $\hat{p}_u = \frac{\exp[m_1(anchor - m_2)]}{\exp[m_1(anchor - m_2)] + \sum_{anchor}}$,
 5:     Compute $m_1$ and $m_2$ according to Eq. 4 and Eq. 6;
      $m_1 = log(\frac{\epsilon \ \hat{p}_u}{(1-\epsilon)(1-\hat{p}_u)})/(anchor - 1), \ m_2 = anchor + log(\frac{1-\hat{p}_u}{\hat{p}_u \sum_{anchor}})/m_1$;
 6:     Update $\hat{p}_i(I)$ and $\mathcal{L}_{ada}$;
      $\hat{p}_i(I) = \frac{\exp[m_1(s(I,T_i) - m_2)]}{\exp[m_1(s(I,T_i) - m_2)] + \sum_{j=1, j\neq i}^{M+1} \exp[s(I,T_j)]}, \ \mathcal{L}_{ada} = \mathbb{E}_{I \sim D}\left[\mathbb{H}(\mathbf{y}(I), \hat{\mathbf{p}}(I))\right]$.
 7: **end for**

---

### 3.4 WEAKLY-SUPERVISED MATCHING WITH PSEUDO CAPTIONS

Existing image-text datasets typically consist of detailed text descriptions to cover fine-grained representations that appear in the corresponding images. However, in practical applications, image-text pairs are mainly scraped from the Internet. These raw datasets not only contain inherent noises but also exhibit a ***high degree of correlation***: for two semantic-distinct images, their corresponding text descriptions may be semantic-consistent. This intrinsic characteristic has been largely overlooked in existing benchmarks. Moreover, research trend has recently been shifting towards prompt-based semi-automatic annotations, which totally differs from current benchmarks.

To validate the applicability of AdaCL in ***a more general annotation setting***, we conduct weakly-supervised image-text matching on AdaCL by annotating images with external tools. Specifically, to ensure the generalizability and tool-agnostic nature of AdaCL, we employ four off-the-shelf image captioning methods (BLIP(Li et al., 2022a), GIT(Wang et al., 2022b), BLIP-2(Li et al., 2023b), and CoCa(Yu et al., 2022)) to generate textual description for images in Flickr30K training set, termed

Table 1: Comparisons of image-text matching. Baselines with three types of backbone are compared, i.e., *ResNet+BiGRU:* CMPM (Zhang & Lu, 2018), VSE++(Faghri et al., 2017); *Faster R-CNN+BiGRU:* SCAN(Lee et al., 2018), IMRAM(Chen et al., 2020a), CVSE(Wang et al., 2020a), SGRAFDiao et al. (2021), CHANPan et al. (2023b); *Faster R-CNN+BERT:* DIME(Qu et al., 2021), CHAN(Pan et al., 2023b), CORA(Pham et al., 2024), USER(Zhang et al., 2024a)

| Methods | MS-COCO (5-fold 1K) | | | | | | MS-COCO (5K) | | | | | | Flickr30K | | | | | |
|---|---|---|---|---|---|---|---|---|---|---|---|---|---|---|---|---|---|---|
| | Image→Text | | | Text→Image | | | Image→Text | | | Text→Image | | | Image→Text | | | Text→Image | | |
| | R@1 | R@5 | R@10 | R@1 | R@5 | R@10 | R@1 | R@5 | R@10 | R@1 | R@5 | R@10 | R@1 | R@5 | R@10 | R@1 | R@5 | R@10 |
| *ResNet +BiGRU* | | | | | | | | | | | | | | | | | | |
| CMPM | 56.1 | 86.3 | 92.9 | 44.6 | 78.8 | 89.0 | 31.1 | 60.7 | 73.9 | 22.9 | 50.2 | 63.8 | 49.6 | 76.8 | 86.1 | 37.3 | 65.7 | 75.5 |
| **AdaCL-CMPM** | **60.7** | **88.2** | **95.0** | **48.5** | **82.8** | **91.4** | **33.9** | **62.3** | **74.4** | **29.1** | **53.3** | **65.5** | **54.7** | **79.0** | **87.5** | **41.6** | **69.4** | **79.2** |
| VSE++ | 64.6 | 90.0 | 95.7 | 52.0 | 84.3 | 92.0 | 41.3 | 71.1 | 81.2 | 30.3 | 59.4 | 72.4 | 52.9 | 80.5 | 87.2 | 39.6 | 70.1 | 79.5 |
| **AdaCL-VSE++** | **65.8** | **91.2** | **96.1** | **54.8** | **85.3** | **93.3** | **44.2** | **72.8** | **83.7** | **33.6** | **61.1** | **73.4** | **55.0** | **84.0** | **87.9** | **44.2** | **73.1** | **81.2** |
| *Faster R-CNN +BiGRU* | | | | | | | | | | | | | | | | | | |
| SCAN | 72.7 | 94.8 | 98.4 | 58.8 | 88.4 | 94.8 | 50.4 | 82.2 | 90.0 | 38.6 | 69.3 | 80.4 | 67.4 | 90.3 | 95.8 | 48.6 | 77.7 | 85.2 |
| **AdaCL-SCAN** | **74.3** | **95.4** | **98.5** | **61.2** | **90.1** | **95.5** | **52.2** | **83.6** | **90.9** | **40.6** | **71.7** | **82.2** | **71.4** | **93.0** | **97.2** | **50.9** | **79.9** | **86.8** |
| IMRAM | 76.7 | 95.6 | 98.5 | 61.7 | 89.1 | 95.0 | 53.7 | 83.2 | 91.0 | 39.7 | 69.1 | 79.8 | 74.1 | 93.0 | 96.6 | 53.9 | 79.4 | 87.2 |
| **AdaCL-IMRAM** | **78.3** | **96.3** | **98.5** | **62.5** | **89.8** | **96.0** | **54.3** | **83.8** | **92.3** | **41.4** | **72.3** | **82.6** | **76.1** | **94.3** | **96.9** | **57.2** | **80.8** | **88.1** |
| CVSE | 78.6 | 95.0 | 97.5 | 66.3 | 91.8 | 96.3 | - | - | - | - | - | - | 73.6 | 90.4 | 94.4 | 56.1 | 83.2 | 90.0 |
| **AdaCL-CVSE** | **79.1** | **95.6** | **97.7** | **67.4** | **93.2** | **97.4** | **56.3** | **83.8** | **92.8** | **41.1** | **73.9** | **83.0** | **74.5** | **91.0** | **95.2** | **56.9** | **83.7** | **91.0** |
| SGRAF | 79.6 | 96.2 | 98.5 | 63.2 | 90.7 | 96.1 | 57.8 | - | 91.6 | 41.9 | - | 81.3 | 77.8 | 94.1 | 97.4 | 58.5 | 83.0 | 88.8 |
| **AdaCL-SGRAF** | **80.3** | **96.7** | **98.5** | **64.2** | **91.6** | **96.4** | **56.0** | **85.1** | **92.9** | **42.3** | **74.1** | **82.7** | **78.9** | **95.6** | **97.8** | **58.7** | **83.2** | **89.5** |
| CHAN | 79.7 | 96.7 | 98.7 | 63.8 | 90.4 | 95.8 | 60.2 | 85.9 | 92.4 | 41.7 | 71.5 | 81.7 | 79.7 | 94.5 | 97.3 | 60.2 | 85.3 | 90.7 |
| **AdaCL-CHAN** | **80.9** | **97.1** | **98.7** | **64.7** | **90.6** | **96.0** | **60.7** | **86.4** | **93.4** | **43.0** | **73.3** | **83.0** | **80.8** | **95.1** | **98.0** | **62.1** | **86.0** | **92.3** |
| *Faster R-CNN +BERT* | | | | | | | | | | | | | | | | | | |
| DIME | 78.8 | 96.3 | 98.7 | 64.8 | 91.5 | 96.5 | 59.3 | 85.4 | 91.9 | 43.1 | 73.0 | 83.1 | 81.0 | 95.9 | 98.4 | 63.6 | 88.1 | 93.0 |
| **AdaCL-DIME** | **80.3** | **96.7** | **99.0** | **65.5** | **91.7** | **96.9** | **60.0** | **85.7** | **92.2** | **44.3** | **73.6** | **83.8** | **82.6** | **96.3** | **98.9** | **63.6** | **88.4** | **93.7** |
| CHAN | 81.4 | 96.9 | 98.9 | 66.5 | 92.1 | 96.7 | 59.8 | 87.2 | 93.3 | 44.9 | 74.5 | 84.2 | 80.6 | 96.1 | 97.8 | 63.9 | 87.5 | 92.6 |
| **AdaCL-CHAN** | **82.1** | **97.2** | **98.9** | **67.9** | **92.5** | **97.5** | **61.1** | **87.9** | **93.3** | **46.0** | **75.0** | **85.6** | **82.0** | **96.5** | **98.2** | **65.6** | **88.3** | **93.3** |
| CORA | 82.8 | 97.3 | 99.0 | 67.3 | 92.4 | 96.9 | 64.3 | 87.5 | 93.6 | 45.4 | 74.7 | 84.6 | 83.4 | 95.9 | 98.6 | 64.1 | 88.1 | 93.1 |
| **AdaCL-CORA** | **83.3** | **97.3** | **99.2** | **66.9** | **92.6** | **97.2** | **67.2** | **88.8** | **94.2** | **47.4** | **76.1** | **86.6** | **83.9** | **95.9** | **98.6** | **64.7** | **89.1** | **93.6** |
| USER | 83.7 | 96.7 | 99.0 | 67.8 | 91.2 | 95.8 | 67.6 | 88.4 | 93.5 | 47.7 | 75.1 | 83.7 | 86.3 | 97.6 | 99.4 | 69.5 | 91.0 | 94.4 |
| **AdaCL-USER** | **84.9** | **97.1** | **99.1** | **67.6** | **91.5** | **95.7** | **68.1** | **88.2** | **94.1** | **47.3** | **76.7** | **83.7** | **86.4** | **98.2** | **99.4** | **69.9** | **90.8** | **94.6** |

as *pseudo captions*. The expression of pseudo captions is structured and standardized (typically in a subject-verb-object order), where more potential clone negatives are included. Then we replace the original annotations with pseudo captions and train with AdaCL separately. Despite the fact that some pseudo captions may not perfectly match their corresponding images, we refrain from re-correcting or modifying them, to ensure a weakly-supervised training manner. In this way, AdaCL can be validated on a coarsely annotated dataset, maximizing the resemblance to image-text pairs collected from the Internet. We demonstrate the results of BLIP as image captioning tool in Section 4.3. Due to the page limit, the complete results of GIT, CoCa, BLIP-2, and the format of pseudo captions are detailed in A.1 and A.2 of the appendix.

# 4 EXPERIMENTS

## 4.1 SETUP

**Datasets.** We evaluate AdaCL on two image-text matching datasets, (1) Flickr30K(Young et al., 2014) consists of 31,783 images, with a training/test/validation split of 29,783/1,000/1,000. (2) MS-COCO(Lin et al., 2014) consists of 123,287 images, with a training/test/validation split of 113,287/5,000/5,000. The test sets are divided into MS-COCO 5-fold 1K (average results of 5 test sets) and MS-COCO 5K (results of 5000 test images).

**Architecture.** For image-text matching, we use three types of backbones: (1) ResNet(He et al., 2016)+BiGRU(Chung et al., 2014), (2) Faster R-CNN(Ren et al., 2015)+BiGRU, and (3) Faster R-CNN+BERT(Devlin et al., 2018). For pre-training, we use CLIP (ViT-B/32), where the embedding size is 512.

**Training Details.** All experiments are performed on four NVIDIA Tesla V100s. For image-text matching, we use a mini-batch size of 64 and the Adam optimizer. The learning rate is 0.0002 and starts decaying 15% of every 10 epochs after epoch 20. The maximum length of each sentence is $a = 32$. For Faster R-CNN, the region number is $n = 36$. The dimension of joint embedding space $D$ is set to 256. We follow (He et al., 2020) to use the momentum memory bank, where the momentum coefficient $z$ is set to 0.99, and the size $M$ is 4096. For AdaCL, $\hat{p}_u$ is set to 0.03, and $\epsilon = e^{-7}$. $m_1$ and $m_2$ are initialized to 20 and 0.1 respectively for adaptive tuning.

**Evaluation Protocols.** Following previous works(Lee et al., 2018), we adopt Recall@K (R@K, K=1, 5, 10) as the evaluation metric: Given a query text, images are ranked based on their similarity to the

query text, and a search is considered correct if at least one relevant image appears within the top K positions in the ranking.

## 4.2 RESULTS ON IMAGE-TEXT MATCHING

**Results on image-text matching baselines.** We choose various competitive image-text matching baselines, and the plug-and-play effectiveness of AdaCL is denoted as "AdaCL-Baseline". From Table 1, it is observed that AdaCL outperforms the original baselines across each backbone type. For MS-COCO, AdaCL-CMPM achieves absolute improvement of (4.6%, 1.9%, 2.1%) on (R@1, R@5, R@10) for image-text matching. For Flickr30K, AdaCL-SCAN achieves absolute improvement of (4.0%, 2.7%, 1.4%) and (2.3%, 2.2%, 1.6%) on (R@1, R@5, R@10) for image-text and text-image matching, demonstrating that AdaCL is applicable to a more general network architecture without elaborate cross-modal fusion designing. In A.5 and A.6, we demonstrate AdaCL's domain generalization capacity in text-based person retrieval, and its robustness in noisy correspondence learning. In A.9, the convergence efficiency of AdaCL is analyzed in detail to better verify its robustness.

**Results on CLIP pre-training.** To assess the scalability, we extend the evaluation to pre-training of a widely used contrastive learning framework CLIP, under two specific large-scale datasets Conceptual Caption 3M (CC3M) and 12M (CC12M) (Sharma et al., 2018). We make a fair comparison by training with 32 epochs under the same experimental settings, and validate the zero-shot image-text retrieval on Flickr30K and MSCOCO. As illustrated in Table 2, AdaCL demonstrates significant improvements under both pre-training datasets, indicating the potential in large-scale contrastive learning framework. In A.7 and A.8 of the appendix, we also report results of AdaCL in zero-shot image classification, and more fine-tuning performance of several other vision-language pre-training methods.

Table 2: Zero-shot retrieval on Flickr30K and MSCOCO. "Baseline" and "AdaCL" represent "CLIP+vanilla CL" and "CLIP+AdaCL".

| Data | Methods | Image→Text | | | Text→Image | | |
|---|---|---|---|---|---|---|---|
| | | R@1 | R@5 | R@10 | R@1 | R@5 | R@10 |
| | | *(Flickr30K)* | | | | | |
| CC3M | Baseline | 26.6 | 52.5 | 63.2 | 18.1 | 39.4 | 49.7 |
| | **AdaCL** | 39.5 | 60.8 | 73.7 | 25.5 | 46.9 | 54.3 |
| CC12M | Baseline | 49.3 | 77.3 | 85.0 | 35.5 | 61.8 | 71.6 |
| | **AdaCL** | 51.0 | 77.5 | 87.9 | 38.4 | 64.6 | 74.7 |
| | | *(MSCOCO)* | | | | | |
| CC3M | Baseline | 13.4 | 32.0 | 43.3 | 10.1 | 25.6 | 35.7 |
| | **AdaCL** | 22.5 | 47.1 | 60.7 | 17.8 | 31.6 | 39.5 |
| CC12M | Baseline | 29.3 | 54.4 | 65.3 | 19.0 | 41.0 | 52.5 |
| | **AdaCL** | 34.0 | 55.6 | 65.9 | 25.1 | 47.3 | 57.4 |

Table 3: Effectiveness of AdaCL under weakly-supervised matching (marked by "PC").

| Methods | Image→Text | | | Text→Image | | |
|---|---|---|---|---|---|---|
| | R@1 | R@5 | R@10 | R@1 | R@5 | R@10 |
| AdaCL-CMPM | 54.7 | 79.0 | 87.5 | 41.6 | 69.4 | 79.2 |
| CMPM (PC) | 42.1 | 68.8 | 82.5 | 31.6 | 57.9 | 70.2 |
| **AdaCL-CMPM (PC)** | 46.6 | 73.4 | 85.2 | 34.3 | 63.5 | 74.1 |
| AdaCL-SCAN | 71.4 | 93.0 | 97.2 | 50.9 | 79.9 | 86.8 |
| SCAN (PC) | 55.4 | 80.2 | 90.6 | 36.9 | 69.7 | 81.6 |
| **AdaCL-SCAN (PC)** | 59.7 | 86.3 | 94.6 | 41.4 | 73.6 | 84.0 |
| AdaCL-CVSE | 74.5 | 91.0 | 95.2 | 56.9 | 83.7 | 91.0 |
| CVSE (PC) | 58.0 | 80.4 | 87.8 | 41.5 | 69.4 | 81.3 |
| **AdaCL-CVSE (PC)** | 64.9 | 84.4 | 92.6 | 47.0 | 76.4 | 86.2 |
| AdaCL-DIME | 82.6 | 96.3 | 98.9 | 63.6 | 88.4 | 93.7 |
| DIME (PC) | 62.1 | 86.7 | 93.0 | 44.7 | 72.9 | 84.5 |
| **AdaCL-DIME (PC)** | 70.8 | 88.3 | 93.6 | 48.7 | 81.8 | 90.4 |

Table 4: Study of components. "PC" and "GT" represent trained with pseudo captions (weakly-supervised), and with ground truth annotations.

| | Modules | | | | PC | GT | Image→Text | | Text→Image | |
|---|---|---|---|---|---|---|---|---|---|---|
| | TRL | CL | AdaCL | Anchor | | | R@1 | R@5 | R@1 | R@5 |
| 1 | ✓ | | | | ✓ | - | 51.4 | 77.3 | 40.5 | 66.8 |
| 2 | | ✓ | | | ✓ | - | 43.7 | 69.2 | 31.0 | 64.7 |
| 3 | | | ✓ | | ✓ | - | 57.2 | 82.5 | 41.4 | 73.4 |
| 4 | | | ✓ | ✓ | ✓ | - | 59.6 | 85.2 | 43.1 | 75.6 |
| 5 | ✓ | | | | - | ✓ | 63.8 | 87.0 | 49.1 | 77.3 |
| 6 | | ✓ | | | - | ✓ | 53.7 | 77.0 | 41.3 | 70.4 |
| 7 | | | ✓ | | - | ✓ | 69.0 | 88.4 | 48.8 | 78.0 |
| 8 | | | ✓ | ✓ | - | ✓ | **74.2** | **91.7** | **53.7** | **81.1** |

Table 5: Comparisons of different anchor selection methods.

| Selection methods (on $\Delta S$) | Image→Text | | Text→Image | |
|---|---|---|---|---|
| | R@1 | R@5 | R@1 | R@5 |
| Random | 64.2 | 85.9 | 44.5 | 75.1 |
| Maximum | 62.7 | 85.1 | 42.7 | 72.9 |
| Minimum | 71.8 | 90.9 | 50.7 | 79.1 |
| **Ours (Median)** | **74.2** | **91.7** | **53.8** | **81.1** |

## 4.3 RESULTS ON WEAKLY-SUPERVISED IMAGE-TEXT MATCHING WITH PSEUDO CAPTIONS

To further validate the applicability of AdaCL in a general annotation setting, we conduct weakly-supervised learning on Flickr30K by replacing the text annotations of the training set with pseudo captions generated by BLIP(Li et al., 2022a). Table 3 demonstrate the performance of AdaCL on several baselines. It is observed that R@10 of AdaCL (PC) is able to rival the original baselines, indicating that AdaCL effectively distinguishes clone negatives and exhibits strong adaptability to

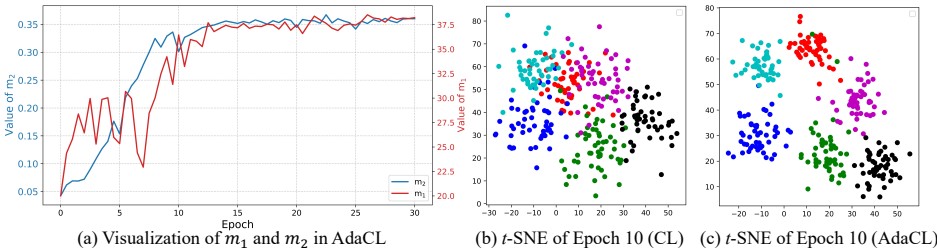

(a) Visualization of $m_1$ and $m_2$ in AdaCL   (b) $t$-SNE of Epoch 10 (CL)   (c) $t$-SNE of Epoch 10 (AdaCL)

Figure 3: Analysis of the tuning process and comparison between AdaCL and CL.

automatically annotated text descriptions. The matching capability of AdaCL under an automated annotation setting is in hot pursuit of the fine-grained human annotations, highlighting the potential of AdaCL in weakly-supervised learning. In A.1 of the appendix, we also report results on three more caption tools.

## 4.4 ABLATION STUDY

In this section, we conduct ablation experiments on Flickr30K to verify AdaCL. For fair comparison, we choose Faster R-CNN + BiGRU for feature extractors, and cross attention for alignment learning in all the ablation studies.

**Verification of modules.** In Table 4, we provide detailed verification by setting different learning objectives during training. "AdaCL" in the $3^{rd}$ column refers to employing AdaCL with a random anchor selection method, and "Anchor" in $4^{th}$ column refers to employing both AdaCL and our proposed anchor selection. Meanwhile, two annotation settings, i.e., ground-truth text annotations (GT) and pseudo captions (PC) are compared. The $1^{st}$ to $4^{th}$ rows showcase the results of pseudo captions (weakly-supervised), and $5^{th}$ to $8^{th}$ rows are results based on the ground truth. AdaCL surpasses both TRL and CL across all metrics by a significant margin, proving that AdaCL further facilitates the learning of cross-modal semantics by adaptive tuning. The proposed anchor selection provides AdaCL with absolute R@1 boost of 5.2%(i-t) and 4.9%(t-i). TRL generally exhibits better performance than vanilla CL, suggesting that TRL is supposed to fit in well with retrieval tasks better than vanilla CL. Meanwhile, we conduct a comprehensive ablation study on other hyper-parameters including momentum memory bank and batch size. Due to page limit, they are demonstrated in A.10.

**Analysis of anchor selection methods.** To verify the rationality of the proposed anchor selection, we compare with several canonical selection methods. The role of *anchor* is to guide the tuning process of $m_1$ and $m_2$ by propagating the semantic of clone negatives. However, as shown in Table 5, selecting the maximum $\Delta S$ contradicts our motivation since its *anchor* has the lowest likelihood of being representative of clone negatives, which naturally leads to poor matching results. On the contrary, minimum selection aligns well with intuition, while it overlooks the possibility of erroneous reasoning by the model itself. Therefore, an intermediary *anchor*, which reflects the average logit of clone negatives, is proved to be the most effective. In addition, we provide an in-depth analysis of the additional arguments involved in anchor selection methodology in A.11 of the appendix.

**Analysis of adaptive tuning process.** Figure 3(a) illustrates the change of the $m_1$ and $m_2$ in AdaCL. During training, $m_2$ gradually increases and stabilizes around 0.36, while $m_1$ exhibits some oscillations before reaching a stable value of approximately 38.1 around Epoch 10, which exhibits a rapid convergence. Figure 3(b) and (c) showcase the $t$-SNE of the fused embedding in Epoch 10. AdaCL exhibits a better clustering performance compared to CL, demonstrating its effectiveness in boosting the convergence of the model during training.

## 4.5 QUALITATIVE ANALYSIS AND VISUALIZATION

**Analysis of clone negatives.** To analyze how AdaCL handles clone negatives, we rank the matching results based on similarity logits and compared three distinct learning objectives. As illustrated in Figure 4, the case study reveals that: (1) AdaCL demonstrates superior R@1 compared to CL and TRL, successfully eliminating other potential negatives; (2) The learned representations of AdaCL exhibit better performance in identifying clone negatives. For instance, the similarity logit for "A

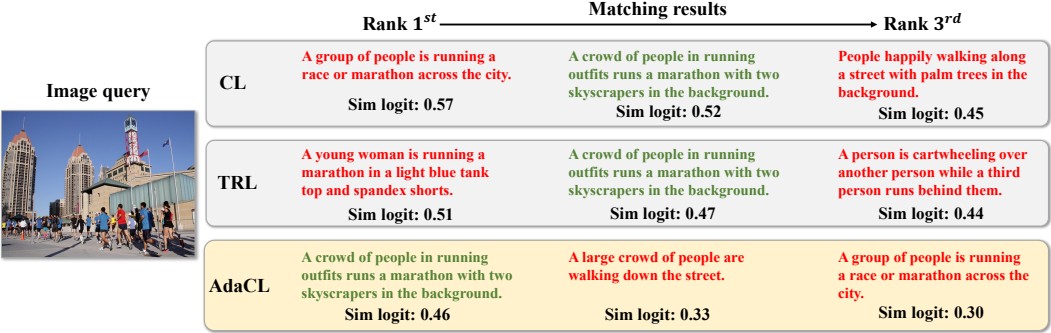

Figure 4: Matching results and similarity logits of clone negatives under CL, TRL, and AdaCL.

group of people running a race or marathon across the city" is merely 0.3. In contrast, both CL and TRL learn sub-optimal similarity logits for clone negatives, despite the presence of the ground-truth.

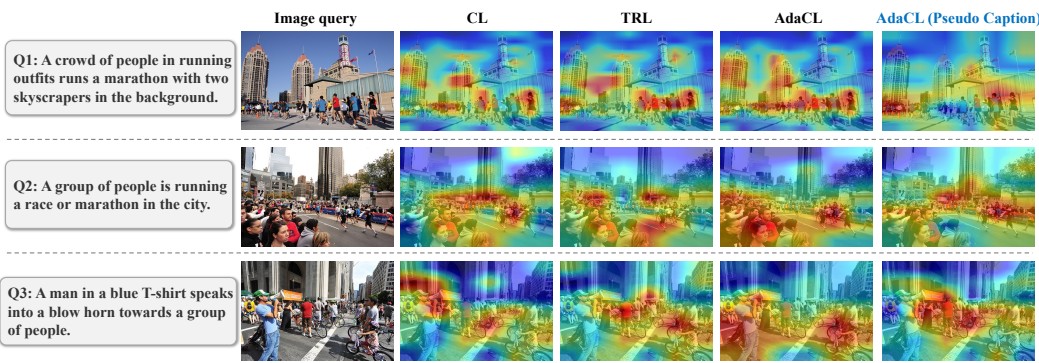

Figure 5: Attention maps in early training stage (Epoch 10). Blue represents AdaCL trained with pseudo captions.

**Visualizations.** We further make qualitative analysis by visualizing several cases in the early training stage to verify the effectiveness of AdaCL in capturing visual cues among clone negatives. From Figure 5, AdaCL are capable of exploring region-level instances together with some background information, such as "two skyscrapers" in Q1 and "blow horn" in Q3. Unexpectedly, we find that the distinguishable features among the clone negatives are also learned by AdaCL. For instance, the de facto distinction between Case 2 and Case 1 is the presence of audiences in Case 2. Even though there is no related key word in Q2 such as "audience" or "spectator", AdaCL still precisely captures the spectators in the lower-left corner of the image. Meanwhile, despite the limited textual contents provided by the pseudo captions, AdaCL roughly capture the presence of the crowd in Case 3, whose effectiveness is slightly better than TRL at the same training stage. Please refer to A.12 of the appendix for more visualization and analysis.

## 5 CONCLUSION

Clone negatives are common but challenging in image-text matching. To mitigate this challenge, we propose Adaptive Contrastive Learning (AdaCL) that introduces two adaptive margins and a modulating anchor to dynamically adjust the compactness between sample pairs and propagates the semantics of clone negatives. The modulating anchor is selected based on the distribution of negative samples without explicit training, allowing for progressive tuning and enhanced in-batch supervision. Furthermore, We extend AdaCL to a weakly-supervised image-text matching by substituting human-annotations with automatically generated captions, increasing the number of potential clone negatives. AdaCL demonstrates robustness in both supervised and weakly-supervised image-text matching. Its superiority demonstrates the potential in alleviating the reliance on crowd-sourced annotations and lays a foundation for vision-language contrastive learning.

ACKNOWLEDGMENTS

This research was partly supported by grants of National Natural Science Foundation of China (NSFC, Grant No. 62171281), Science and Technology Commission of Shanghai Municipality (STCSM, Grant Nos. 20DZ1200203, 2021SHZDZX0102).

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

# A APPENDIX

The appendix is structured as follows:
§A.1 provides experimental results on weakly-supervised image-text matching with pseudo captions generated by GIT, CoCa, and BLIP-2.
§A.2 demonstrates the expression format of pseudo captions generated by the image captioning tools.
§A.3 provides the detailed derivation of anchor selection.
§A.4 provides the detailed derivation of Equation 9.
§A.5 provides experimental results on text-based person retrieval.
§A.6 provides experimental results on noisy correspondence learning.
§A.7 provides experimental results on zero-shot image classification of CLIP pre-training.
§A.8 provides experimental results on VLP fine-tuning.
§A.9 provides efficiency analysis of AdaCL.
§A.10 provides a comprehensive study of other hyper-parameters.
§A.11 provides an analysis on additional arguments involved in anchor selection.
§A.12 provides a more comprehensive visualization results of clone negatives.
§A.13 provides a discussion of the limitations for this work.

## A.1 WEAKLY-SUPERVISED IMAGE-TEXT MATCHING

In the main manuscript, we utilize BLIP(Li et al., 2022a) to generate pseudo captions based on Flickr30K training set. Then, the original text annotations are replaced by the pseudo captions for training to verify the robustness of AdaCL in handling clone negatives. To mitigate concerns about reliance on a specific captioning tool, we conduct a range of complementary experiments, to comprehensively analyze its robustness. Specifically, four captioning tools are selected, i.e., GIT(Wang et al., 2022b), CoCa(Yu et al., 2022), and BLIP-2(Li et al., 2023b). GIT is a multi-modal pre-training method that unifies vision-language tasks such as image/video captioning and question answering. CoCa employs a unified transformer architecture to perform both image-text matching and image captioning tasks. CoCa is trained on large-scale image-text pairs and can generate descriptive captions for images while also understanding the relationship between visual and textual content. BLIP-2 is an advanced vision-language model that builds upon its predecessor, BLIP. It introduces a lightweight Querying Transformer (Q-Former) to bridge pre-trained vision and language models efficiently.

The way of generating pseudo captions is unified, i.e., through the zero-shot image captioning results. The maximum length of the pseudo caption is 30. Table 6 demonstrates the matching results of different baselines with AdaCL based on pseudo captions. The experimental settings are the same with the settings in the main manuscript.

From Table 6, we can make the following conclusions: By employing different image captioning method, AdaCL demonstrates matching performance that are within a **3%** margin. AdaCL achieves highly competitive performance on the four annotation settings. Therefore, AdaCL is further proved to be applicable to more general annotation settings, where the issue of clone negatives is well mitigated. The impact of image captioning methods on AdaCL is trivial. More specifically, AdaCL-$\mathcal{X}$ (BLIP) and AdaCL-$\mathcal{X}$ (BLIP-2) performs slightly better than AdaCL-$\mathcal{X}$ (GIT) and AdaCL-$\mathcal{X}$ (CoCa). We speculate that this is because the pseudo captions generated by GIT and CoCa typically focus on action and instance information of the image gallery, while the pseudo captions generated by BLIP possess general descriptions, often consisting of a subject-verb-object structure with more holistic-level semantics. Overall, the retrieval performance is quite comparable, further verifying the robustness of AdaCL.

## A.2 EXPRESSION FORMAT OF PSEUDO CAPTIONS

Regarding pseudo captions, we have presented a subset of caption cases generated by four distinct captioning tools (BLIP, GIT, BLIP-2, and CoCa), as illustrated in Figure 6. Pseudo captions generally provide a global and coarse overview of the images, encompassing more potential clone negatives. Among the four methods, GIT produces the most concise captions, while CoCa tends to generate more detailed descriptions. The weakly-supervised image-text matching based on pseudo captions

Table 6: Comparisons of Image-Text Retrieval performance on Flickr30K test set with pseudo captions generated by four distinct captioning methods. AdaCL-$\mathcal{X}$ (BLIP), AdaCL-$\mathcal{X}$ (GIT), AdaCL-$\mathcal{X}$ (CoCa), and AdaCL-$\mathcal{X}$ (BLIP-2) represent AdaCL with respective image captioning methods. **Bold** is the best performance, while red indicates the margin between the best and worst.

| Methods | Image→Text | | | Text→Image | | |
|---|---|---|---|---|---|---|
| | R@1 | R@5 | R@10 | R@1 | R@5 | R@10 |
| AdaCL-CMPM (BLIP) | **46.3** | 72.9 | **85.8** | 34.1 | 63.6 | 75.0 |
| AdaCL-CMPM (GIT) | 44.5 (-1.8) | 71.2 | 84.9 (-0.9) | 32.7 (-1.8) | 61.0 (-2.7) | 73.7 (-1.5) |
| AdaCL-CMPM (CoCa) | 44.8 | 71.0 (-2.4) | 85.1 | 34.1 | **63.7** | 74.6 |
| AdaCL-CMPM (BLIP-2) | 45.9 | **73.4** | 85.7 | **34.5** | 63.3 | **75.2** |
| AdaCL-SCAN (BLIP) | 59.2 | 86.9 | 94.7 | 41.7 | **73.2** | 84.1 |
| AdaCL-SCAN (GIT) | **60.0** | **87.7** | **94.9** | 41.1 | 70.9 (-2.3) | 83.4 (-1.6) |
| AdaCL-SCAN (CoCa) | 58.1 (-1.9) | 85.2 | 94.2 (-0.7) | 40.2 (-2.0) | 72.6 | 83.6 |
| AdaCL-SCAN (BLIP-2) | 58.1 | 85.2 (-2.5) | 94.6 | **42.2** | 74.1 | **85.0** |
| AdaCL-CVSE (BLIP) | 64.7 | 82.6 | **92.9** | **47.0** | 77.5 | 88.4 |
| AdaCL-CVSE (GIT) | 63.6 (-1.9) | 80.9 (-1.8) | 90.9 (-2.0) | 46.2 | 77.4 | **88.7** |
| AdaCL-CVSE (CoCa) | 63.8 | 81.4 | 91.5 | 45.3 (-1.7) | 77.1 (-0.6) | 88.0 (-0.7) |
| AdaCL-CVSE (BLIP-2) | **65.5** | **82.7** | 92.3 | 46.3 | **77.7** | 88.2 |
| AdaCL-DIME (BLIP) | **71.3** | **88.3** | **94.9** | 54.7 | 82.8 | 90.4 |
| AdaCL-DIME (GIT) | 70.1 (-1.2) | 88.0 | 94.2 | 55.1 | 82.3 | 90.6 |
| AdaCL-DIME (CoCa) | 70.8 | 87.8 (-0.5) | 93.7 | 54.6 (-1.3) | 82.0 (-1.4) | 90.0 (-1.0) |
| AdaCL-DIME (BLIP-2) | 70.4 | **88.3** | 93.7 (-1.2) | **55.9** | **83.4** | **91.0** |

poses imposes demands for handling clone negatives, thus rendering this task more challenging. We will release all datasets based on pseudo captions to facilitate further research in this domain.

### A.3 DERIVATION OF $m_1$ AND $m_2$ IN ADACL

Revisiting AdaCL, **our goal is to progressively tune $m_1$ and $m_2$ based on the *anchor*.** Therefore, we first copy Equation 3 in the main manuscript, i.e., softmax normalized similarity for each image $I$ and its corresponding text $T$ with two margin parameters, which can be expressed as:

$$\hat{p}_i(I) = \frac{\exp\left[m_1(s(I, T_i) - m_2)\right]}{\exp\left[m_1(s(I, T_i) - m_2)\right] + \sum_{j-1, j\neq i}^{M+1} \exp\left[s(I, T_j)\right]}, \tag{12}$$

$$\mathcal{L}_{ada} = \mathbb{E}_{I\sim D}\left[\mathbb{H}(\mathbf{y}(I), \mathbf{p}(I))\right] = -\frac{1}{N}\sum_{i=1}^{N} y_i(I) \log(\hat{p}_i(I)). \tag{13}$$

The potential in-batch clone negatives are represented as: $S^* := \{s \mid p(\mathcal{C} \mid s) > p(\bar{\mathcal{C}} \mid s)\}$, and *anchor* is defined as the median of $S^*$. The two specific boundary functions of the anchor are defined as:

$$\hat{p}_u = \frac{\exp\left[m_1(anchor - m_2)\right]}{\exp\left[m_1(anchor - m_2)\right] + \sum_{anchor}}, \tag{14}$$

$$\frac{\exp\left[m_1(1 - m_2)\right]}{\exp\left[m_1(1 - m_2)\right] + \sum_{anchor}} = 1 - \epsilon, \tag{15}$$

where $\sum_{anchor}$ is the simplification of $\sum_{k=1, k\neq u}^{M+1} \exp\left[s(I_u, T_k)\right]$. $\sum_{anchor}$ can be obtained through Equation 14, expressed as:

$$\sum\nolimits_{anchor} = \frac{1-\hat{p}_u}{\hat{p}_u} e^{m_1(anchor - m_2)}. \tag{16}$$

Combining Equation 16 and Equation 15, we have:

$$\left[e^{m_1(1-m_2)} + \frac{1 - \hat{p}_u}{\hat{p}_u} e^{m_1(anchor - m_2)}\right] (1 - \epsilon) = e^{m_1(1-m_2)}. \tag{17}$$

| Image Query | Ground-truth | BLIP | GIT | BLIP-2 | CoCa |
|---|---|---|---|---|---|
| | A band is playing to a cheering concert with many people. | A crowd of people in a concert with a band on stage. | Many people are enjoying a concert. | A crowd of people in a large concert. | A large crowd of people are gathered in a concert. |
| | Music being played by several individuals while a crowd sits and listens. | People sitting in chairs watching a man play a musical instrument. | People having an indoor concert. | A group of people and a man playing a violin | A group of musicians sitting in a room with instruments . |
| | Two men who are riding on a horse both are trying to rope a bull in a rodeo. | There is a man riding a horse with a cow. | A man riding a horse chasing the cow. | A man is on a horse | A man on a horse roping a calf in a rodeo. |
| | A man wearing blue jeans is trying to stop a horse. | A man is falling off of a horse. | A man is riding a horse. | A man is trying to catch a horse that is running away. | A man in a black and white striped shirt is trying to rope a horse . |
| | A crowded sidewalk in the inner city of an Asian country. | People walking on a street in a city with shops. | People walking on the street. | A group of people walking down a street | A group of people walking down a street . |
| | A crowd of people is walking down the middle of a city street. | There are many people walking down the street together. | Many people gather on the street. | A large crowd of people walking down a street. | A crowd of people walking down a street . |
| | A crowd of people in running outfits runs a marathon with two skyscrapers in the background. | People running in a marathon in a city with tall buildings. | People are jogging during the day. | A group of people running in a city | A group of people walking on the sidewalk near a building . |
| | A group of people is running a race or marathon in the city. | People are running in a marathon in a city street | Athletes running in the city. | A large crowd of people running in a marathon | A group of people that are standing in the street . |
| | A man in a blue T-shirt speaks into a blow horn towards a group of people. | People are walking in a crowded city street. | Cyclists riding across the street. | A man is standing on a street corner with a megaphone | A group of people riding bikes down a street . |
| | Many people are chilling in front an old building. | People are in front of a large building with a clock tower. | Tourists sitting at tables outside the building. | A crowd of people sitting at tables outside of a building. | A crowd of people sitting at tables in front of a building. |
| | A group of people stand in the park of a city, with buildings in the background. | People standing around a fountain in a city with tall buildings | Several people are standing in the park. | A group of people standing in the park. | A group of people standing in front of a water fountain . |

Figure 6: Cases of pseudo captions by four distinct captioning tools, i.e., BLIP, GIT, BLIP-2, and CoCa.

To simplify Equation 17, we have:

$$\frac{(1-\epsilon)(1-\hat{p}_u)}{\hat{p}_u} e^{m_1 \cdot anchor} = \epsilon \cdot e^{m_1}. \tag{18}$$

By taking the logarithm of both sides of Equation 18, we have:

$$m_1 \cdot anchor + log\frac{(1-\epsilon)(1-\hat{p}_u)}{\hat{p}_u} = m_1 + log\ \epsilon. \tag{19}$$

Then, $m_1$ can be obtained, expressed as:

$$m_1 = log(\frac{\epsilon\ \hat{p}_u}{(1-\epsilon)(1-\hat{p}_u)})/(anchor - 1), \tag{20}$$

which corresponds to Algorithm 1 in the manuscript. Meanwhile, by taking the logarithm of both sides of Equation 16, we have:

$$log\sum\nolimits_{anchor} = log\frac{1-\hat{p}_u}{\hat{p}_u} + m_1(anchor - m_2). \tag{21}$$

Simplifying Equation 21, we obtain $m_2$:

$$m_2 = anchor + log(\frac{1-\hat{p}_u}{\hat{p}_u \cdot \sum_{anchor}})/m_1, \tag{22}$$

which corresponds to Algorithm 1 in the main manuscript. With Equation 20 and Equation 22, $m_1$ and $m_2$ can be computed and updated during each batch training process with the supervision of *anchor*, facilitating the model to exploit more distinguishable cross-modal semantics among samples compared with the original TRL and CL.

## A.4 DERIVATION OF EQUATION 9

Here we demonstrate the derivation of Equation 9. To begin with, a $K$-class classification probability with Bayes's formula can be expressed as:

$$p(y = i \mid x) = \frac{p(x \mid y = i)p(y = i)}{\sum_{j=1}^{K} p(x \mid y = j)p(y = j)} = \frac{\exp\left(f_i(x)\right)}{\sum_{j=1}^{K} \exp\left(f_j(x)\right)}, \tag{23}$$

In anchor selection of AdaCL, the input variable $x$ (a.k.a $s$) is one-dimensional with a binary output variable $y \in 0, 1$ (a.k.a $\bar{\mathcal{C}}$ and $\mathcal{C}$). We aim to predict $p(y = 1 \mid x)$. Since GDA assumes that for each class $y = 0$ and $y = 1$, the input $x$ follows a gaussian distribution. This can be expressed as: $p(x \mid y = 0) = \mathcal{N}\left(x \mid \mu_0, \sigma_0^2\right)$ and $p(x \mid y = 1) = \mathcal{N}\left(x \mid \mu_1, \sigma_1^2\right)$. $\mu_0$, $\mu_1$ and $\sigma_0^2$, $\sigma_1^2$ are the means and variances of distributions for classes $y = 0$ and $y = 1$, respectively. Thus, the posterior probability can be expressed as:

$$p(y = 0 \mid x) = \frac{\mathcal{N}\left(x \mid \mu_0, \sigma_0^2\right) \cdot \pi_0}{\mathcal{N}\left(x \mid \mu_0, \sigma_0^2\right) \cdot \pi_0 + \mathcal{N}\left(x \mid \mu_1, \sigma_1^2\right) \cdot \pi_1}, \tag{24}$$

$$p(y = 1 \mid x) = \frac{\mathcal{N}\left(x \mid \mu_1, \sigma_1^2\right) \cdot \pi_1}{\mathcal{N}\left(x \mid \mu_1, \sigma_1^2\right) \cdot \pi_1 + \mathcal{N}\left(x \mid \mu_0, \sigma_0^2\right) \cdot \pi_0}. \tag{25}$$

Since $\mathcal{N}\left(x \mid \mu_1, \sigma_1^2\right)$ is the probability density function of a Gaussian distribution, we substitute $\mathcal{N}\left(x \mid \mu_1, \sigma_1^2\right) = \frac{1}{\sqrt{2\pi\sigma^2}} \exp\left(-\frac{(x-\mu)^2}{2\sigma^2}\right)$ into the above equations, obtaining Equation 9 in the manuscript:

$$p\left(y = 0 \mid x\right) = \frac{1}{1 + \frac{\pi_1}{\pi_0}\frac{\sigma_0}{\sigma_1}\exp\left[\frac{(s-\mu_0)^2}{2\sigma_0^2} - \frac{(s-\mu_1)^2}{2\sigma_1^2}\right]}, \tag{26}$$

$$p\left(y = 1 \mid x\right) = \frac{1}{1 + \frac{\pi_0}{\pi_1}\frac{\sigma_1}{\sigma_0}\exp\left[\frac{(s-\mu_1)^2}{2\sigma_1^2} - \frac{(s-\mu_0)^2}{2\sigma_0^2}\right]}, \tag{27}$$

$p\left(y = 0 \mid x\right)$ and $p\left(y = 1 \mid x\right)$ represent the probability of a similarity score to be a clone negative or not, without the need of explicit pre-processing to the dataset or training.

Table 7: R@1 Results on text-based person search. "DG" stands for domain generalization, and "FT" for fine-tuning on the corresponding dataset.

| Method | CUHK-PEDES | | ICFG-PEDES | | RSTPReid | |
|--------|------|------|------|------|------|------|
| | DG | FT | DG | FT | DG | FT |
| CL | 16.3 | 49.3 | 15.8 | 43.5 | 12.7 | 30.1 |
| **AdaCL** | **30.5** | **56.7** | **27.9** | **49.0** | **23.9** | **41.4** |

## A.5 DOMAIN GENERALIZATION ON TEXT-BASED PERSON RETRIEVAL

To evaluate the robustness of AdaCL as a plug-and-play module, we seek to evaluate its domain generalization capabilities in text-based person retrieval. Specifically, we conduct training using Flickr30K and select three mainstream text-based person retrieval datasets, CUHK-PEDES(Li et al., 2017b), ICFG-PEDES(Ding et al., 2021), and RSTPReid(Zhu et al., 2021) for domain generalization experiments. To ensure a fair comparison, we choose CMPM as the baseline, as the learning objective it adopt in its paper is the closest to vanilla contrastive learning, and its original paper indeed conducted experiments on two of the datasets. As illustrated in Table 7, it is observed that AdaCL boosts CMPM by a large margin. Especially for DG, the results of AdaCL on each dataset improves by over 10%.

Table 8: Fine-tuning results of AdaCL on three baselines under ICFG-PEDES.

| Method | Text-Image R@1 | Text-Image R@5 | Text-Image R@10 |
|--------|------|------|------|
| CMPM | 43.5 | 65.4 | 74.2 |
| **AdaCL-CMPM** | **49.0** | **69.7** | **79.1** |
| ViTAA | 51.0 | 68.8 | 75.8 |
| **AdaCL-ViTAA** | **54.8** | **74.1** | **78.6** |
| IRRA | 63.5 | 80.3 | 85.8 |
| **AdaCL-IRRA** | **64.3** | **81.1** | **86.5** |

Table 9: Fine-tuning results of AdaCL on three baselines under RSTPReid.

| Method | Text-Image R@1 | Text-Image R@5 | Text-Image R@10 |
|--------|------|------|------|
| CMPM | 30.1 | 38.5 | 59.6 |
| **AdaCL-CMPM** | **41.4** | **57.0** | **55.7** |
| ViTAA | 37.7 | 60.6 | 66.5 |
| **AdaCL-ViTAA** | **42.6** | **62.1** | **69.2** |
| IRRA | 60.2 | 81.3 | 88.2 |
| **AdaCL-IRRA** | **62.7** | **81.4** | **89.0** |

For fine-tuning, in addition to CUHK-PEDES, we also validate the performance of AdaCL on ICFG-PEDES and RSTPReid. Three baselines are employed: CMPM(Zhang & Lu, 2018), ViTAA(Wang et al., 2020b), and IRRA(Jiang & Ye, 2023), and compare the effectiveness of incorporating AdaCL as a constraint. The experimental results w/ and w/o using AdaCL are presented in Table 8 and Table 9. It is observed that AdaCL demonstrates significant improvements across the three baselines, achieving absolute enhancements of 5.5%, 3.8%, and 0.8% in R@1, respectively. These matching results substantiate the robustness of AdaCL in other vision-language downstream tasks, demonstrating its insensitivity to the diverse dataset distributions (both natural images and person search images), and the choice of baselines.

## A.6 NOISY CORRESPONDENCE LEARNING

As mentioned in Section 1, noisy correspondence learning (NC) (Huang et al., 2021; Yang et al., 2023; Ma et al., 2024; Qin et al., 2023) focuses on handling negatives by manually introducing noisy labels. Several works classify samples into clean and noisy subsets, followed by a rectifier and triplet

ranking loss to boost the learning of NC. We further validate AdaCL in such challenging scenarios by plugging in AdaCL and verify its NC effectiveness on Flickr30K using the same pre-processing strategy (by shuffling the captions of training images for a specific percentage, denoted by noise ratio). The matching results under two noise ratio (20% and 40%) are reported in Table 10.

Table 10: Noisy correspondence learning of AdaCL. We follow (Huang et al., 2021) to shuffle the captions of training images for a specific percentage, i.e., noise ratio.

| Noise Ratio | Methods | Image→Text | | | Text→Image | | |
|---|---|---|---|---|---|---|---|
| | | R@1 | R@5 | R@10 | R@1 | R@5 | R@10 |
| 20% | NCR | 75.0 | 93.9 | 97.5 | 58.3 | 83.0 | 89.0 |
| | **AdaCL-NCR** | 75.3 | 93.8 | 97.4 | 61.2 | 84.1 | 89.7 |
| | BiCro | 78.1 | 94.4 | 97.5 | 60.4 | 84.4 | 89.9 |
| | **AdaCL-BiCro** | 79.6 | 95.2 | 97.5 | 62.7 | 85.1 | 91.3 |
| | CREAM | 77.4 | 95.0 | 97.3 | 58.7 | 84.1 | 89.8 |
| | **AdaCL-CREAM** | 80.0 | 95.6 | 97.4 | 61.9 | 86.4 | 91.3 |
| | CRCL | 77.9 | 95.4 | 98.3 | 60.9 | 84.7 | 90.6 |
| | **AdaCL-CRCL** | 81.0 | 96.2 | 98.5 | 62.3 | 84.9 | 91.7 |
| 40% | NCR | 68.1 | 89.6 | 94.8 | 51.4 | 78.4 | 84.8 |
| | **AdaCL-NCR** | 74.7 | 92.3 | 96.6 | 57.8 | 82.0 | 87.1 |
| | BiCro | 74.6 | 92.7 | 96.2 | 55.5 | 81.1 | 87.4 |
| | **AdaCL-BiCro** | 75.3 | 93.1 | 96.2 | 57.4 | 82.5 | 89.6 |
| | CREAM | 76.3 | 93.4 | 97.1 | 57.0 | 82.6 | 88.7 |
| | **AdaCL-CREAM** | 79.2 | 95.1 | 98.3 | 61.5 | 86.0 | 90.2 |
| | CRCL | 77.8 | 95.2 | 98.0 | 60.0 | 84.0 | 90.2 |
| | **AdaCL-CRCL** | 80.3 | 95.0 | 98.1 | 61.7 | 84.4 | 90.9 |

We also validate the effectiveness of AdaCL on CC152K. CC152K consists of 150,000 samples from training split of Conceptual Captions (CC) (Sharma et al., 2018) for training, 1,000 samples from validation split for validation, and 1,000 samples from validation split for testing. As all image-text pairs in CC are automatically harvested from the Internet, approximately 3%–20% of the pairs in the dataset are mismatched or weakly matched. This benchmark aligns well with the settings of NC, making it a suitable choice for evaluating AdaCL.

From Table 10 and Table 11, it can be concluded that for a noise ratio of 20%, AdaCL achieves notable improvements, particularly in I-T R@1 (AdaCL-CRCL improves from 77.9 to 81.0) and T-I R@1 (AdaCL-NCR improves from 58.3 to 61.2). For a noise ratio of 40%, the trend of improvement remains consistent, although the performance naturally decreases as noise increases. Notably, AdaCL-CRCL demonstrates strong robustness with I-T R@1 improving from 77.8 to 80.3, even at high noise levels. While the baseline results degrade significantly as the noise ratio increases, AdaCL exhibits better resilience, as evidenced in AdaCL-NCR (I-T R@1 only drops from 75.3 to 74.7). AdaCL's robustness is particularly evident in T-I matching, where the decline in performance is less pronounced compared to the baselines (AdaCL-CRCL achieves T-I R@5 of 84.4 at 40% noise ratio). Similar to Flickr30K, AdaCL also demonstrates consistent improvements over the baselines on CC152K. The performance improvements of AdaCL on both datasets further support its generalizability and applicability in noisy correspondence learning.

Table 11: Noisy correspondence learning of AdaCL on CC152K.

| Methods | Image→Text | | | Text→Image | | |
|---|---|---|---|---|---|---|
| | R@1 | R@5 | R@10 | R@1 | R@5 | R@10 |
| NCR | 39.5 | 64.5 | 73.5 | 40.3 | 64.6 | 73.2 |
| **AdaCL-NCR** | 43.2 | 66.9 | 74.9 | 42.5 | 69.0 | 76.2 |
| BiCro | 40.8 | 67.2 | 76.1 | 42.1 | 67.6 | 76.4 |
| **AdaCL-BiCro** | 42.9 | 66.1 | 76.0 | 42.7 | 68.4 | 78.7 |
| CREAM | 40.3 | 68.5 | 77.1 | 40.2 | 68.2 | 78.3 |
| **AdaCL-CREAM** | 43.1 | 69.6 | 77.2 | 42.2 | 70.0 | 80.2 |
| CRCL | 41.8 | 67.4 | 76.5 | 41.6 | 68.0 | 78.4 |
| **AdaCL-CRCL** | 42.4 | 68.0 | 77.4 | 41.7 | 69.3 | 80.0 |

## A.7 Zero-shot Image Classification of AdaCL in CLIP pre-training

In addition to image-text matching, we also evaluate AdaCL on other pre-training task, i.e., zero-shot image classification. Specifically, we validate AdaCL on eight common classification benchmarks, which can be divided into (i) general datasets: ImageNet(Deng et al., 2009), CIFAR-10(Krizhevsky et al., 2009), CIFAR-100(Krizhevsky et al., 2009), Caltech-101(Fei-Fei et al., 2004)), and (ii) fine-grained datasets: Food-101(Bossard et al., 2014), Flowers-102(Nilsback & Zisserman, 2008), OxfordPets(Parkhi et al., 2012), and FGVCAircraft(Maji et al., 2013). The Top-1 accuracy results of "CLIP + AdaCL" pretrained on CC3M and CC12M are demonstrated in Table 12:

Table 12: Zero-shot image classification of CLIP pre-training under different learning objectives. "Baseline" represents "CLIP+vanilla contrastive learning", and "AdaCL" represents "CLIP+AdaCL". Results under two pre-training settings, i.e., CC3M and CC12M are compared.

| Data | Model | Datasets | | | | | | | |
|------|-------|----------|---|---|---|---|---|---|---|
| | | ImageNet | CIFAR-10 | CIFAR-100 | Caltech-101 | Food-101 | Flowers | Pets | Aircraft |
| CC3M | Baseline | 17.2 | 71.3 | 32.1 | 50.9 | 10.2 | 10.8 | 12.1 | 1.0 |
| | **AdaCL** | **22.0** | **77.1** | **42.2** | **54.8** | **12.6** | **13.3** | **14.9** | **1.7** |
| CC12M | Baseline | 32.9 | 72.5 | 38.0 | 74.0 | 26.5 | **25.7** | 46.2 | 2.6 |
| | **AdaCL** | **34.8** | **73.4** | **43.3** | **74.7** | **33.1** | 25.4 | **46.7** | **2.8** |

It is observed that AdaCL outperforms CL in all the general datasets and most of the fine-grained datasets, proving its advantage in recognition tasks. Specifically, in ImageNet, CIFAR-10, CIFAR-100, the Top-1 accuracy of AdaCL has surpassed vanilla CL by over 5%. It is noteworthy that the performance on fine-grained datasets further verifies AdaCL's capacity in challenging scenarios.

## A.8 VLP Fine-tuning

Apart from CLIP pre-training, we further report the fine-tuning results of AdaCL in several Vision Language Pre-training methods (VLP) by fine-tuning them using AdaCL on MS-COCO (5K). As illustrated in Table 13, AdaCL facilitates matching performance across nearly all metrics under both dual-encoder method (BEIT-3(Wang et al., 2022c)) and fusion-encoder methods (UNITER(Chen et al., 2020c), OSCAR(Li et al., 2020), VinVL(Zhang et al., 2021)), effectively boosting the fine-tuning process. These results further corroborate the robustness of AdaCL across multiple baselines.

Table 13: Results of AdaCL on VLP fine-tuning.

| Methods | Image→Text | | | Text→Image | | |
|---------|------|------|------|------|------|------|
| | R@1 | R@5 | R@10 | R@1 | R@5 | R@10 |
| UNITER† | 65.7 | 88.6 | 93.8 | 52.9 | 79.9 | 88.0 |
| **AdaCL-UNITER** | **67.6** | **89.0** | **94.3** | **55.1** | **81.2** | **88.9** |
| OSCAR | 70.0 | 91.1 | 95.5 | 54.0 | 80.8 | 88.5 |
| **AdaCL-OSCAR** | **71.0** | **92.7** | **96.3** | 54.0 | 80.6 | **89.1** |
| VinVL | 75.4 | 92.9 | 96.2 | 58.8 | 83.5 | 90.3 |
| **AdaCL-VinVL** | **78.7** | **94.4** | **96.8** | **60.4** | **84.2** | **91.1** |
| BEIT-3 | 84.8 | 96.5 | 98.3 | 67.2 | 87.7 | 92.8 |
| **AdaCL-BEIT-3** | 84.4 | **96.9** | 98.3 | **68.6** | **89.1** | **93.7** |

† Evaluated by us with official repository.

## A.9 Efficiency Analysis

Serving as a plug-and-play module, AdaCL does not increase the inference time since it is independent of the cross-modal reasoning module. For training efficiency, we add detailed analysis on AdaCL.

The speed of model convergence mirrors the learning efficiency of a certain constraint. As shown in Figure 7, AdaCL brings considerable convergence efficiency and retrieval results, which even boost CMPM and DIME to achieve their ultimate results within the first 5 epochs, demonstrating the scalability and efficiency of AdaCL.

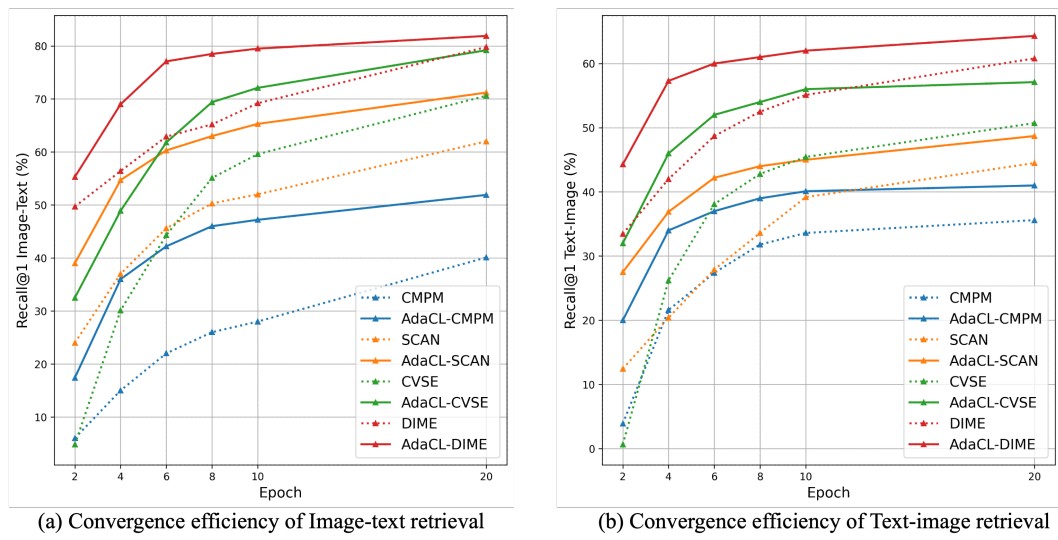

(a) Convergence efficiency of Image-text retrieval          (b) Convergence efficiency of Text-image retrieval

Figure 7: Training efficiency of AdaCL.

## A.10  ABLATION STUDIES OF OTHER HYPER-PARAMETERS

### A.10.1  ANALYSIS OF MOMENTUM MEMORY BANK

Since memory bank is widely adopted in vision-language contrastive learning, we further analyze AdaCL by varying the memory bank size $M$, which leads to different number of negative samples. The results in Table 14 reveal that the momentum memory bank yields a modest yet discernible improvement: Among the sizes of $4096$, $6144$, and $8192$, the impact of memory bank is not that significant. This suggests that AdaCL does not excessively rely on the quantity of negative samples for an ideal similarity distribution.

Table 14: Effect of different memory bank sizes.

| Memory Bank Size | Image→Text | | | Text→Image | | |
|---|---|---|---|---|---|---|
| | R@1 | R@5 | R@10 | R@1 | R@5 | R@10 |
| N/A | 70.3 | 90.0 | 95.5 | 49.5 | 77.5 | 87.3 |
| 2048 | 71.9 | 90.9 | 96.7 | 51.4 | 78.4 | 87.3 |
| 4096 | **74.2** | 91.7 | 97.9 | **53.7** | 81.1 | **88.2** |
| 6144 | 73.7 | **91.9** | **98.2** | 53.2 | 79.6 | 87.8 |
| 8192 | 73.9 | 91.4 | 98.0 | 53.3 | **80.5** | 87.8 |

### A.10.2  ANALYSIS OF MINI-BATCH SIZE

Since mini-batch size is correlated with the number of potential anchor candidates for selection, we also investigate the impact of mini-batch size, as shown in Table 15. It can be observed that AdaCL exhibits remarkable robustness to variations in batch size settings. Across a range of batch sizes from 16 to 128, the fluctuation in R@1 remains within a narrow margin of 4%.

Table 15: Effect of different mini-batch sizes.

| Batch Size | Image→Text | | | Text→Image | | |
|---|---|---|---|---|---|---|
| | R@1 | R@5 | R@10 | R@1 | R@5 | R@10 |
| 16 | 70.6 | 89.2 | 96.5 | 50.0 | 79.3 | 84.8 |
| 32 | 72.9 | 90.9 | 97.4 | 52.1 | 81.0 | 87.1 |
| 64 | **74.2** | **91.7** | **97.9** | 53.7 | 81.1 | **88.2** |
| 128 | 74.0 | 91.2 | 97.6 | **53.9** | **81.2** | 88.0 |

## A.11 ADDITIONAL PARAMETERS IN ANCHOR SELECTION

Here we further analyze the anchor selection methodology. In the main manuscript, we utilize the negatives from $S_{sln}$ and $S_{cln}$ within each mini-batch to represent the empirical means and variances. However, we cannot guarantee that all samples in $S_{sln}$ and $S_{cln}$ are exclusively salient negatives and clone negatives. Based on this speculation, we continue to select Top-$K$ similarity scores from $S_{sln}$ and $S_{cln}$ as observational samples for calculating means and variances. The assumption of this study is higher similarity scores for certain negatives correlate with an increased probability of them being clone negatives. We set $K$ to 32 and conduct experiments on CMPM, SCAN, and DIME under Flickr30K, as shown in Table 16. It can be concluded that employing Top-$K$ selection strategy does not result in a significant improvement or deterioration in matching performance, with fluctuations generally remaining within a 2% range. This observation contradicts our initial hypothesis and intuition. Consequently, we can infer that AdaCL exhibits ***low sensitivity to the specific values of the empirical mean and variance***, which is another minor merit. Given that the Top-$K$ selection explicitly increase computation without yielding significant performance improvements, we have opted to maintain the original calculation method in the main manuscript.

Table 16: Matching results of Top-$K$ selection for empirical means and variances.

| Methods | Image→Text | | | Text→Image | | |
|---|---|---|---|---|---|---|
| | R@1 | R@5 | R@10 | R@1 | R@5 | R@10 |
| AdaCL-CMPM | 54.7 | 79.0 | 87.5 | 41.6 | 69.4 | 79.2 |
| AdaCL-CMPM † | 54.2 | 77.8 | 87.1 | 41.8 | 68.1 | 80.0 |
| AdaCL-SCAN | 71.4 | 93.0 | 97.2 | 50.9 | 79.9 | 86.8 |
| AdaCL-SCAN † | 72.7 | 93.4 | 96.5 | 50.2 | 79.0 | 87.1 |
| AdaCL-DIME | 82.6 | 96.3 | 98.9 | 63.6 | 88.4 | 93.7 |
| AdaCL-DIME † | 82.4 | 95.3 | 98.7 | 63.7 | 88.2 | 93.0 |

†: Employ Top-$K$ selection.

## A.12 MORE VISUALIZATION OF ADACL

We present a more comprehensive comparison of CL, TRL, and AdaCL trained with ground-truth annotations and pseudo captions. The visualization results of the early training stage are demonstrated in Figure 8, which include 4 kinds of clone negatives with 11 cases. Based on the attention maps, we can summarize the following conclusions: AdaCL captures abundant semantics on highly similar clone negatives. Specifically, case (a) and case (b) demonstrate that AdaCL boosts the exploration of spatial semantics among the images, such as "music being played by several individuals", as well as "is trying to stop a horse", which effectively distinguishes clone negatives apart. Additionally, case (c) demonstrates AdaCL's ability in capturing background information such as "Asian country" and "a city street" are crucial phrases that are reasoned through AdaCL. Case (d) showcases five examples of urban landscape, demonstrating that AdaCL is able to discover instances that are not explicitly described in the text query. For instance, the unique attribute "spectator" is not included in Q8, but AdaCL facilitate learning the corresponding representation, which is highlighted in the attention map. Also, the latent "fountain" is not included in Q11 but reasoned by AdaCL. In this way, AdaCL is proved to achieve comprehensive cross-modal semantics with its adaptive tuning strategy even when

the quality of textual annotations is not high. This finding presents great potential of AdaCL to handle retrieval with low quality labels.

Furthermore, we obtain the attention maps by training with pseudo captions under AdaCL, as depicted in the last column of Figure 8. Due to the lack of instance-level information during the training process, we do not expect the results to surpass models trained on original annotations. However, AdaCL (Pseudo Caption) manages to capture the approximate cross-modal semantics and pays attention to the fine-grained representation, which outperforms CL and TRL (trained with ground-truth) in most cases. This demonstrates the prospects of AdaCL in the vision-language contrastive learning of automatically annotated image-text pairs.

### A.13 DISCUSSION: LIMITATION

In this work, AdaCL is evaluated on (1) image-text matching under Flickr30K, MS-COCO, (2) CLIP pre-training under CC3M and CC12M, (3) weakly-supervised image-text matching under pseudo captions, (4) text-based person search under CUHK-PEDES, ICFG-PEDES, and RSTPReid. We have not extended AdaCL to an all-round vision-language tasks due to time and computational limitations, which is undoubtedly planned in our future endeavor.

Also, although AdaCL maintains high convergence efficiency, we acknowledge that AdaCL inevitably introduces additional computation during training with a moderate computational overhead of $O(N \cdot M)$ per batch training. We believe this trade-off is acceptable given the context of contrastive learning and pre-training. In future work, we will delve into a more lightweight vision-language learning paradigm.

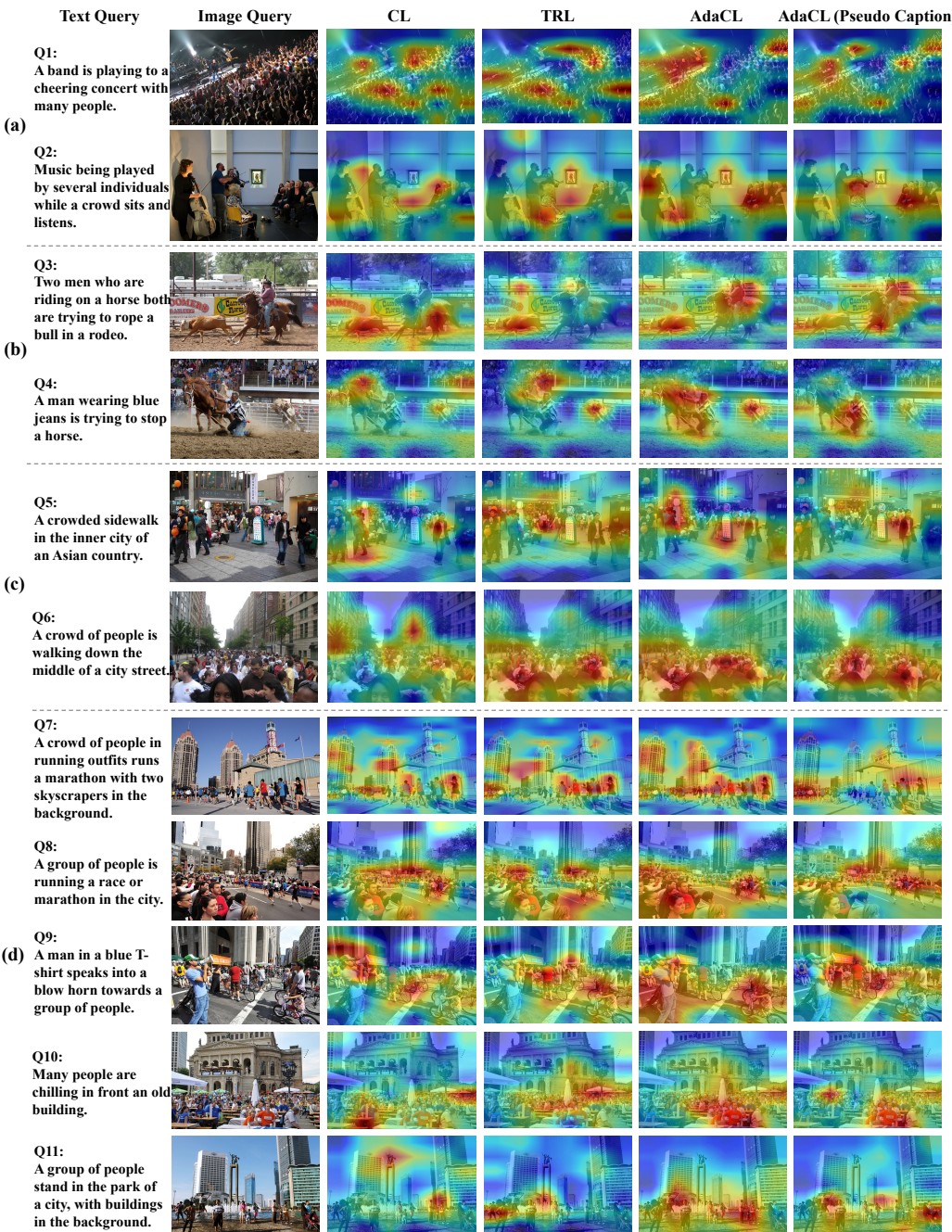

Figure 8: Attention maps of clone negative cases in early stage (Epoch 10). "CL", "TRL", and "AdaCL" represent model trained with different constraints. The last column represents AdaCL trained with pseudo captions.

