# OpenReview forum: "Discovering Clone Negatives via Adaptive Contrastive Learning for Image-Text Matching"
_ICLR.cc/2025/Conference — ICLR 2025 Poster_

### Official Review · Reviewer_8azX · 2024-10-31

**Soundness:** 3
**Presentation:** 3
**Contribution:** 3
**Rating:** 8
**Confidence:** 4

**Summary:**

The authors of this paper propose a new method to improve cross-modal learning by adding a conditioning factor to the loss function to alleviate what the authors call "clone negatives". The proposed method is effective from the experimental results. However, the concept of clone negatives is more like the False-negative problem mentioned in some existing studies. For this reason, the concept of clone negatives still needs further discussion.

**Strengths:**

1. The author's idea of ​solving the false-negative problem is clear.
2. The experiments are rich and the proposed method is effective.

**Weaknesses:**

1. The authors proposed a new concept, namely clone negative examples. However, this is actually the false-negative problem, which is not a new topic. For example, there has been similar research in [1,2].
2. The authors should enrich the caption of Figure 2 to make the core idea clear and concise.
3. Line325: “Comparisons of Image-text matching” should be “Comparisons of image-text matching”
4. The 2024 baselines are missing.
5. Line 954,955: Missing citations.
6. There are usually three mainstream benchmarks for text-based person search tasks. I suggest the author add the results of RSTPReid. Also, does the authors‘ R@ refer to the Recall accuracy? However, in the text-based person search task, the Rank accuracy is usually used as the evaluation indicator.

Reference:

[1] Li H, Bin Y, Liao J, et al. Your negative may not be true negative: Boosting image-text matching with false negative elimination[C]//Proceedings of the 31st ACM International Conference on Multimedia. 2023: 924-934.

[2] Li Z, Guo C, Feng Z, et al. Integrating language guidance into image-text matching for correcting false negatives[J]. IEEE Transactions on Multimedia, 2023, 26: 103-116.

**Questions:**

See weakness.

---

> ### Author Response · Authors · 2024-11-20
> **Official reply to Reviewer 8azX [1/2]**
>
> We appreciate your time and effort in reviewing our paper and providing valuable feedback to help us improve it. Below, we address your comments point by point:
>
>
> >(1) **W1:** *The authors proposed a new concept, namely clone negative examples. However, this is actually the false-negative problem, which is not a new topic. For example, there has been similar research in [1,2].*
>
> Thank you for your feedback regarding the motivation. We would like to clarify that clone negatives differ fundamentally from false negatives. Let us try to analyze them in details:
>
> - False negatives are true positives mislabeled due to annotation or sampling errors, and their impact on contrastive learning is well understood. They are regarded as "true matches" that are incorrectly categorized as negatives during the training process. The solutions to false negatives are usually improved sampling and elimination strategies, as highlighted in the papers mentioned by the reviewer.
>
>
> - Clone negatives present a novel challenge: they are samples that closely resemble the ground-truth positive but miss the **specific and fine-grained details** of the true positive, which is defined as sub-optimal (true negatives). For example, in Fig. 1 of the manuscript, the general concept of a marathon or race exists in $I_3$ (a clone negative), but lacking critical attributes "two skyscrapers in the background" in $I_1$ (ground-truth). Clone negatives occupy an intermediate category that, while sharing high-level semantic similarities with anchor, but fail to achieve a "true" match to the anchor. Rather than being categorized as "false negatives", clone negatives are critical to contrastive learning because they lie betwee true positives and hard negatives, creating ambiguous training signals that require specific handling.
>
>
>
> In addition to the difference in definition, existing methods primarily handle false negatives by applying smaller penalties to them, i.e., through an elimination approach. Specifically, [1] cut down the sampling weights of false negatives to minimize their impact during training. Similarly, [2] filters false negatives through a manually-set fixed threshold, and employs sampling to lower their weights, followed by using a vanilla contrastive loss as the learning objective. Fundamentally, these methods adopt a **"passive"** approach to "avoid" false negatives.
>
>
> On the contrary, AdaCL is proposed to **"actively"** address clone negatives by dynamically adjusting the two introduced margin parameters **within contrastive learning process**, which are vital to the quality of learned representation. In this way, AdaCL facilitates distinguishing clone negatives through the boosts of trainable parameters within each baseline. Therefore, AdaCL is proved to be an effective plug-and-play learning objective in various downstream tasks.
>
>
> Therefore, clone negatives introduced in this paper represents a unique set of samples that standard contrastive learning or image-text matching baselines struggle to handle. In the revised manuscript, we have refined the definition of clone negatives to strengthen the motivation of our work. Also, we cite and compare several related works on false negative elimination for a comprehensive literature review. We hope the above discussion eliminates your concern, and we sincerely thank you for your valuable feedback.
>
> [1] Li H, Bin Y, Liao J, et al. Your negative may not be true negative: Boosting image-text matching with false negative elimination[C]//Proceedings of the 31st ACM International Conference on Multimedia. 2023: 924-934.
>
> [2] Li Z, Guo C, Feng Z, et al. Integrating language guidance into image-text matching for correcting false negatives[J]. IEEE Transactions on Multimedia, 2023, 26: 103-116.
>
>
> >(2) **W2:** *The authors should enrich the caption of Figure 2 to make the core idea clear and concise.*
>
> Thank you for your valuable suggestion. In the revised manuscript, we have enriched the caption for Fig. 2 to provide a more detailed explanation. Specifically, we have emphasized the process of AdaCL and outlined its overall pipeline. We believe this enhancement makes the core idea clearer and more concise, facilitating a better understanding of our paper.
>
>
> >(3) **W3:** *Line325: "Comparisons of Image-text matching" should be "Comparisons of image-text matching".* **W5:** *Line 954,955: Missing citations.*
>
> Thank you for your careful review and valuable feedback. In the revised manuscript, we have corrected all the typos and checked all the hyperlinks and citations to improve the overall quality.

---

> ### Author Response · Authors · 2024-11-20
> **Official reply to Reviewer 8azX [2/2]**
>
> >(4) **W4:** *The 2024 baselines are missing.*
>
> In response to the reviewer's suggestion regarding the baselines, we have further verified AdaCL on two more recent state-of-the-art image-text baselines CORA [3] and USER [4]. We followed the same experimental settings in the manuscript, and their respective plug-and-play effectiveness is demonstrated as follows:
>
>
> |   Method   |  Dataset  | I-T R@1 | I-T R@5 | I-T R@10 | T-I R@1 | T-I R@5 | T-I R@10 |
> |:----------:|:---------:|:-------:|:-------:|:--------:|:-------:|:-------:|:--------:|
> | AdaCL-CORA | MSCOCO-1K | 83.3 (82.8)  | 97.3 (97.3)  |  99.2 (99.0)  | 66.9 (67.3)  | 92.6 (92.4)  |  97.2 (96.9)  |
> | AdaCL-USER | MSCOCO-1K | 84.9 (83.7)  | 97.1 (96.7)  |  99.1 (99.0)  | 67.6 (67.8)  | 91.5 (91.2)  |  95.7 (95.8)  |
> | AdaCL-CORA | MSCOCO-5K | 67.2 (64.3)  | 88.8 (87.5)  |  94.2 (93.6)  | 47.4 (45.4)  | 76.1 (74.7)  |  86.6 (84.6)  |
> | AdaCL-USER | MSCOCO-5K | 68.1 (67.6)  | 88.2 (88.4)  |  94.1 (93.5)  | 47.3 (47.7)  | 76.7 (75.1)  |  83.7 (83.7)  |
> | AdaCL-CORA | Flickr30K | 83.9 (83.4)  | 95.9 (95.9)  |  98.6 (98.6)  |  64.7 (64.1)  | 89.1 (88.1)  |  93.6 (93.1)  |
> | AdaCL-USER | Flickr30K | 86.4 (86.3)  | 98.2 (97.6)  |  99.4 (99.4)  | 69.9 (69.5)  | 90.8 (91.0)  |  94.6 (94.4)  |
>
> The results in parentheses, i.e., $(\cdot)$ represent results of the original baselines. We can see that most of the two baselines have been improved, further demonstrating the effectiveness of AdaCL in image-text matching. In the revised manuscript, we add the above results to Tab. 1 of Sec. 4, including an all-round set of influential baselines that incorporate diverse visual and textual backbones.
>
> [3] Zhang et al., "USER: Unified semantic enhancement with momentum contrast for image-text retrieval." TIP 2024.
>
> [4] Pham et al., "Composing object relations and attributes for image-text matching." CVPR 2024.
>
>
> >(5) **W6:** *There are usually three mainstream benchmarks for text-based person search tasks. I suggest the author add the results of RSTPReid. Also, does the authors' R@ refer to the Recall accuracy? However, in the text-based person search task, the Rank accuracy is usually used as the evaluation indicator.*
>
> We appreciate the reviewer for your constructive suggestion. We have further validated AdaCL on RSTPReid. The following table demonstrate the Recall@1 of AdaCL. We choose CMPM as baseline for a fair comparison, because the learning objective it adopt in its paper is the closest to vanilla CL. "DG" stands for the domain generalization results (trained on Flickr30K and zero-shot on three text-based person search datasets). "FT" represents fine-tuning on the corresponding datasets. The result on RSTPReid is evaluated by us with CMPM's official repository.
>
>
> | Method | CUHK-PEDES (DG) | CUHK-PEDES (FT) | ICFG-PEDES (DG) | ICFG-PEDES (FT) | RSTPReid (DG)    | RSTPReid (FT)    |
> | ------ | --------------- | --------------- | --------------- | --------------- | --- | --- |
> | CL     | 16.3            | 49.3            | 15.8            | 43.5            | 12.7    | 30.1    |
> | **AdaCL**  | 30.5        | 56.7        | 27.9        | 49.0        | 23.9    | 41.4    |
>
> It is observed that AdaCL demonstrates the robustness in both domain generalization and fine-tuning, especially for DG, the results of AdaCL on each dataset improves by over 10%.
>
> For fine-tuning, in addition to CUHK-PEDES and ICFG-PEDES in the original manuscript, we also validate AdaCL on RSTPReid with three baselines. The results are demonstrated as follows:
>
> | Method          | Text-Image R@1 | Text-Image R@5 | Text-Image R@10 |
> | --------------- | -------------- | -------------- | --------------- |
> | CMPM            | 30.1           |  48.5          | 59.6            |
> | **AdaCL-CMPM**  | **41.4**       | **57.0**       | **55.7**        |
> | ViTAA           | 37.7           | 60.6           | 66.5            |
> | **AdaCL-ViTAA** | **42.6**       | **62.1**       | **69.2**        |
> | IRRA            | 60.2           | 81.3           | 88.2            |
> | **AdaCL-IRRA**  | **62.7**       | **81.4**       | **89.0**        |
>
> We can see that our AdaCL is insensitive to diverse dataset distributions (both natural images and person search images). All the above results have been added to the appendix of the revised manuscript for a comprehensive comparison.
>
> For evaluation metric, R@K refers to Recall@K. Specifically, given a query text, images are ranked based on their similarity to the query text. A search is considered correct if at least one relevant image appears within the top K positions in the ranking. To the best of our knowledge, this evaluation metric is widely used in text-based person search. Recall@K is indeed the evaluation metric we used in the experiments. In the revised manuscript, we have clarified the notation in the relevant sections to eliminate any potential ambiguity. We hope this addresses your concern and makes the evaluation metric clearer.

---

> ### Author Response · Authors · 2024-11-24
>
> Dear reviewer 8azX,
>
> Thank you so much for your time and efforts in reviewing our paper. We have addressed your comments in detail and are happy to discuss more if there are any additional concerns. We are looking forward to your feedback and would greatly appreciate you consider raising the scores.
>
> Best regards,
>
> Authors

---

> > ### Comment · Reviewer_8azX · 2024-11-25
> >
> > Thank you for your detailed reply. I still have some questions. 1. The concepts of Recall and RanK are different, one is to calculate the ratio, and the other is to calculate the hit. 2. Some recent works in image-text matching for challenging settings need to be discussed, such as noisy correspondence [1,2,3,4]. I wonder if AdaCL can handle this kind of noise with its unique adaptive mechanism. I look forward to the author's further reply. I will keep the score for now.
> >
> > [1] Learning with noisy correspondence for cross-modal matching, NIPS 21.\
> > [2]  Cross-modal active complementary learning with self-refining correspondence, NIPS 23.\
> > [3] Bicro: Noisy correspondence rectification for multi-modality data via bi-directional cross-modal similarity consistency, CVPR 23.\
> > [4] UGNCL: uncertainty-guided noisy correspondence learning for efficient cross-modal matching, SIGIR 24.

---

> > > ### Author Response · Authors · 2024-11-26
> > > **Follow-up reply to Reviewer 8azX [1/2]**
> > >
> > > Thank you for your valuable feedback. Below we address your questions point-by-point.
> > >
> > > **(i).** Recall@K (R@K) is indeed the evaluation metric we use in the experiments, which indicates the percentage of queries for which relevant results were successfully retrieved. For each query, if at least one relevant result is retrieved among the top-K results, the query is considered a successful retrieval. Otherwise, it is recorded as a failed retrieval.
> > >
> > > In cross-modal retrieval [1][2] and text-based person retrieval [3] tasks, Recall@K is commonly used as an evaluation metric. Its calculation method is identical to that of Rank-K in pedestrian re-identification [4] tasks.
> > >
> > > It is worth noting that in search and recommendation systems, Recall@K has a different definition. In such contexts, it measures the proportion of relevant results retrieved within the top K results out of all relevant results available for the query [5].
> > >
> > >
> > > [1] Stacked cross attention for image-text matching, Proceedings of the European conference on computer vision, ECCV 2018.
> > >
> > > [2] Focus your attention: A bidirectional focal attention network for image-text matching, ACM MM 2019.
> > >
> > > [3] Vitaa: Visual-textual attributes alignment in person search by natural language, ECCV 2020.
> > >
> > > [4] Local fisher discriminant analysis for pedestrian re-identification, CVPR 2013.
> > >
> > > [5] https://amitness.com/posts/information-retrieval-evaluation

---

> > > ### Author Response · Authors · 2024-11-26
> > > **Follow-up reply to Reviewer 8azX [2/2]**
> > >
> > > **(ii).** Noisy Correspondence Learning (NC) is a challenging scenario for image-text matching, which refers to the alignment errors in paired data. We further validate AdaCL in such a challenging scenario on Flickr30K using the same dataset shuffling strategy as the related baselines (shuffling the captions of training images for a specific percentage, denoted by noise ratio). Specifically, we adopt a series of NC methods and replace their label rectification process and learning objective with AdaCL during training. The matching results under two noise ratio (20% and 40%) are reported as follows, where the results in parentheses, i.e., $(\cdot)$ represent the baseline performance.
> > >
> > > |    Noise ratio   |   Method    |   I-T R@1   |   I-T R@5   |  I-T R@10   |   T-I R@1   |   T-I R@5   |  T-I R@10   |
> > > |:-----------:|:-----------:|:-----------:|:-----------:|:-----------:|:-----------:|:-----------:|:-----------:|
> > > |     20%     |  AdaCL-NCR   |   75.3 (75.0)    |   93.8 (93.9)    |   97.4 (97.5)    |   61.2 (58.3)    |   84.1 (83.0)    |   89.7 (89.0)    |
> > > |     20%     | AdaCL-BiCro |   79.6 (78.1)    |   95.2 (94.4)    |   97.5 (97.5)    |   62.7 (60.4)    |   85.1 (84.4)    |   91.3 (89.9)    |
> > > |     20%     | AdaCL-CREAM |   80.0 (77.4)    |   95.6 (95.0)    |   97.4 (97.3)    |   61.9 (58.7)    |   86.4 (84.1)    |   91.3 (89.8)    |
> > > |     20%     | AdaCL-CRCL  |   81.0 (77.9)    |  96.2  (95.4)    |   98.5 (98.3)    |  62.3 (60.9)    |    84.9 (84.7)    |   91.7 (90.6)    |
> > > | ----------- | ----------- | ----------- | ----------- | ----------- | ----------- | ----------- | ----------- |
> > > |     40%     |  AdaCL-NCR  |   74.7 (68.1)    |   92.3 (89.6)    |   96.6 (94.8)    |   57.8 (51.4)    |   82.0 (78.4)    |   87.1 (84.8)    |
> > > |     40%     | AdaCL-BiCro |   75.3 (74.6)    |   93.1 (92.7)    |   96.2 (96.2)    |  57.4 (55.5)    |   82.5 (81.1)    |   89.6 (87.4)    |
> > > |     40%     | AdaCL-CREAM |  79.2 (76.3)    |  95.1 (93.4)    |  98.3 (97.1)    |  61.5 (57.0)    |  86.0 (82.6)    |  90.2 (88.7)    |
> > > |     40%     | AdaCL-CRCL  |   80.3 (77.8)    |  95.0 (95.2)    |  98.1 (98.0)    |  61.7 (60.0)    |  84.4 (84.0)    |  90.9 (90.2)    |
> > >
> > >
> > > Due to the lack of code availability, we are unable to include UGNCL [1] within the current rebuttal timeline. Instead, we incorporated a 2024 work CREAM [2], as an additional baseline. We also validate the effectiveness of AdaCL on the CC152K. CC152K consists of 150,000 samples from training split of Conceptual Captions(CC) for training, 1,000 samples from validation split for validation, and 1,000 samples from validation split for testing. As all image-text pairs in CC are automatically harvested from the Internet, there exists approximately 3%–20% of the pairs in the dataset mismatched or weakly matched. This benchmark aligns well with the challenging scenarios, making it suitable for evaluating AdaCL. We report the results as follows:
> > >
> > >
> > > |   Method    | I-T R@1 | I-T R@5 | I-T R@10 | T-I R@1 | T-I R@5 | T-I R@10 |
> > > |:-----------:|:-------:|:-------:|:--------:|:-------:|:-------:|:--------:|
> > > |  AdaCL-NCR  | 43.2 (39.5)  | 66.9 (64.5)  | 74.9 (73.5)  | 42.5 (40.3)  | 69.0 (64.6)  | 76.2 (73.2)  |
> > > | AdaCL-BiCro | 42.9 (40.8)  | 66.1 (67.2)  | 76.0 (76.1)  | 42.7 (42.1)  | 68.4 (67.6)  | 78.7 (76.4)  |
> > > | AdaCL-CREAM | 43.1 (40.3)  | 69.6 (68.5)  | 77.2 (77.1)  | 42.2 (40.2)  | 70.0 (68.2)  | 80.2 (78.3)  |
> > > | AdaCL-CRCL  |  42.4 (41.8)  | 68.0 (67.4)  | 77.4 (76.5)  |41.7 (41.6)  | 69.3 (68.0)  | 80.0 (78.4)  |
> > >
> > > From the above two tables, we can see that AdaCL consistently outperforms the majority of the baselines, indicating its effectiveness in noisy correspondence learning for image-text matching. For a noise ratio of 20%, AdaCL achieves notable improvements, particularly in I-T R@1 (AdaCL-CRCL improves from 77.9 to 81.0) and T-I R@1 (AdaCL-NCR improves from 58.3 to 61.2). For a noise ratio of 40%, the improvement trend remains consistent, despite that the performance naturally decreases as noise increases. Notably, while the baseline results degrade significantly as the noise ratio increases, AdaCL exhibits better resilience, as evidenced in AdaCL-NCR (I-T R@1 only drops from 75.3 to 74.7). Also, the additional parameters introduced by AdaCL for adaptive tuning are minimal, highlighting its efficiency. Similar to Flickr30K, AdaCL also demonstrates consistent improvements over the baselines on CC152K. The performance improvements of AdaCL on both datasets further support its generalizability and applicability in noisy correspondence learning.
> > >
> > >
> > > All the above results, along with detailed analysis and references to related literature, will be included in the revised manuscript to further enhance the completeness of this paper if necessary.
> > >
> > >
> > > [1] UGNCL: Uncertainty-Guided Noisy Correspondence Learning for Efficient Cross-Modal Matching, SIGIR 2024.
> > >
> > > [2] Cross-modal Retrieval with Noisy Correspondence via Consistency Refining and Mining, TIP 2024.

---

> ### Comment · Reviewer_8azX · 2024-11-26
>
> Thanks for your reply, I'll raise my score to 8.

---

> > ### Author Response · Authors · 2024-11-26
> >
> > We sincerely appreciate your positive feedback and support for our work, as well as your constructive comments and suggestions, which greatly enhance the quality of our paper.

---

### Official Review · Reviewer_vmnW · 2024-11-02

**Soundness:** 3
**Presentation:** 2
**Contribution:** 3
**Rating:** 6
**Confidence:** 2

**Summary:**

This paper recognises the clone negative issue in image-text matching, where negative image-text pairs are semantically consistent with the positives. To address this issue, the authors modulate the softmax function of original contrastive losses with anchors that are selected from each in-batch similarity score based on Gaussian discriminant analysis. The proposed method, called Adaptive Contrastive Learning, demonstrates superiority in both supervised and weakly-supervised image-text matching.

**Strengths:**

1. The motivation of this work is clear. The issue of clone negatives is well illustrated and intriguing, which is common and significant in matching images and texts.

2. The proposed method is extensively verified under both supervised and weakly supervised settings, exhibiting superior performance under several benchmarks.

3. The authors provide comprehensive ablation studies and analyses, with both quantitative and qualitative verification.

**Weaknesses:**

1. The method is not applied to or compared with the recent CLIP-style models, which I believe would strengthen the significance and soundness of this work.

**Questions:**

1. From what I understand, this method is matching images with texts. Then why is FasterRCNN, an object detection model, used in the experiments?

---

> ### Author Response · Authors · 2024-11-20
> **Official reply to Reviewer vmnW**
>
> Thank you for noting the quality of our work. Here, we reply to your comments:
>
>
> >(1) **W1:** *The method is not applied to or compared with the recent CLIP-style models, which I believe would strengthen the significance and soundness of this work.*
>
> We sincerely appreciate the reviewer's constructive suggestion. In the revised manuscript, we have further included pre-training results for "CLIP + AdaCL" and provided a detailed comparison with "CLIP + Vanilla CL." For further details, please refer to our Meta Reply.
>
>
> >(2) **Q1:** From what I understand, this method is matching images with texts. Then why is FasterRCNN, an object detection model, used in the experiments?
>
> Thankyou for your careful review. We would be happy to clarify this in more detail. As we mentioned in the introduction, there are usually global-level aligning and object-level aligning in image-text matching, with the key difference being the **granularity of matching**. Global-level aligning uses the visual backbone to capture a holistic representation of the image. In contrast, object-level aligning leverages a fine-grained visual representation for matching. Therefore, a great number of methods employ Faster R-CNN to extract the object representation $f_{img} \in \mathbb{R}^{K\times D}$, where $K$ stands for the number of objects within an image. The use of Faster R-CNN enables the models to focus on  aligning objects and entities, eliminating the impact of background noises. Therefore, object detection model is widely used in image-text matching task.

---

> > ### Comment · Reviewer_vmnW · 2024-11-20
> >
> > Thanks for the explanation. I will keep my positive rating.

---

> > > ### Author Response · Authors · 2024-11-23
> > > **Thanks for your reply**
> > >
> > > Dear Reviewer vmnW,
> > >
> > >
> > > We greatly appreciate your positive feedback and support. Thank you again for your reply and for taking time to consider our response.
> > >
> > >
> > > Best, Authors

---

### Official Review · Reviewer_PFrK · 2024-11-02

**Soundness:** 3
**Presentation:** 3
**Contribution:** 3
**Rating:** 8
**Confidence:** 5

**Summary:**

This paper focused on solving the task of Image-Text Mathing (ITM) especially the key challenge of clone negatives: the captions that have consistent semantics with the GT sentences, but with less coarse-grained information. To solve this, the Adaptive Contrasvite Learning (AdaCL) method is proposed, which introduces supervision of the clone negatives, and can dynamically penalize the clone negatives. This will enlarge the distance between the positives and the clone negatives.

**Strengths:**

1. The authors focus on a critical issue of clone negatives in the field of Contrastive Learning under the background of Vision-Language Models and Image-Text Matching Tasks.
2. Experiments are well performed, which show effectiveness of AdaCL.

**Weaknesses:**

1. More pre-trained models and more down-stream tasks could be test to further demonstrate the effectiveness of the proposed method.

2. Spelling error(s):
p2 077 andand
p2 Fig 1(b) T2 -> T3
p1 034 action-level -> global level
p6 274 double curly braces “}” -> just one “}”, delete the last one

3. Sequential adverbs:
p4 205 There are “Two conditions”. I can see the “First”, but where is the “Second”?
It should be before p4 214 ”(Second, )to achieve this”.

4. Meaning of subscript:
p4 208 What does the subscript “u” mean? What is its abbreviation? Is it “unique(==specific)”?
And what about p4 210 the “u-” in k∈M_u-? Is it “not unique”? I can’t sure.

5. Inconsistencie(s):
p4 Sec 3.2 one anchor + one pos + M neg ==> Why not all M neg? Because Sec 3.2 and Sec 3.3 are connected.
p5 Sec 3.3 one anchor + one pos + (M-1) neg
p6 Algorithm 1 a mini-batch of N pos + M neg??? ==> Why not one pos + M neg?
  May be the authors want to say “a mini-batch of N, in which 1 pos and M neg”?
  So, N==1+M?

6. p5 254 what is the superscript of j in Eq. (8)? Is it the same as Eq. (7) or Eq. (3)?

7. p5 263 Why Eq. (9) is that? Can you give us a concrete deduction? And what about p(C=0|s)? Can you give a specific formulation?

8. p10 509 How to correctly understand the heat map, since a sentence contain ALL WORDS, and every mentioned instance in the image SHOULD be highlited. So, what does these Attention Map mean indeed? What Information can I get from them? Or what should we expect from them?

**Questions:**

Please refer to weakness above.

---

> ### Author Response · Authors · 2024-11-20
> **Official reply to Reviewer PFrK [1/2]**
>
> Thank you for summarizing our work and highlighting our motivation and well performed experiments. Below, we provide a point-by-point response to your comments.
>
> >(1) **W1:** *More pre-trained models and more down-stream tasks could be test to further demonstrate the effectiveness of the proposed method.*
>
>
> Thank you for your constructive suggestion. In the revised manuscript, we have extended our evaluation by pre-training AdaCL on CLIP with two large-scale datasets. Therefore, we have validated its effectiveness on (1) image-text matching, (2) weakly-supervised image-text matching (3) text-based person search, (4) zero-shot image-text retrieval (pre-training), and (5) zero-shot image classification (pre-training) tasks. For further details about pre-training, please refer to our Meta Reply.
>
> >(2) **W2:** *Spelling error(s): p2 077 andand p2 Fig 1(b) T2 -> T3 p1 034 action-level -> global level p6 274 double curly braces "}" -> just one "}", delete the last one*
>
> We sincerely appreciate your thorough review. In the revised manuscript, we have meticulously reviewed and corrected all typos to enhance overall quality.
>
> >(3) **W3:** *Sequential adverbs: p4 205 There are “Two conditions”. I can see the “First”, but where is the “Second”? It should be before p4 214 ”(Second, )to achieve this”.*
>
> Your understanding is entirely correct. The second condition represents the boundary analysis of Eq. 4, as defined following the phrase "To achieve this". We apologize for the ambiguity and have refined the expression in this section in the revised manuscript.
>
> >(4) **W4:** *Meaning of subscript: p4 208 What does the subscript “u” mean? What is its abbreviation? Is it “unique(==specific)”? And what about p4 210 the “u-” in k∈M_u-? Is it “not unique”? I can’t sure.*
>
> In line 210, "$u$" denotes to the index of the selected anchor, and "$\bar{u}$" refers to the samples excluding the anchor. We further change the original expression to $\sum_{\textit{anchor}} = \sum_{k=1, k\neq u}^{M+1} {\rm exp} \left[s(I_u, T_k)\right]$ to make it clearer. Meanwhile, retaining "$u$" preserves the intended clarity by indicating that a specific anchor is being referenced. Thus, to improve readability, we have removed the superscript "anc" in Eq. 4, as it does not provide further utility within the equation. We believe this simplification enhances the clarity of the notation. Thank you for your helpful suggestion.
>
> >(5) **W5:** *Inconsistencie(s): p4 Sec 3.2 one anchor + one pos + M neg ==> Why not all M neg? Because Sec 3.2 and Sec 3.3 are connected. p5 Sec 3.3 one anchor + one pos + (M-1) neg p6 Algorithm 1 a mini-batch of N pos + M neg??? ==> Why not one pos + M neg? May be the authors want to say “a mini-batch of N, in which 1 pos and M neg”? So, N == 1+M?*
>
> In p4 Sec 3.2, the analysis considers 1 positive, and its corresponding $M$ negatives. In p5 Sec 3.3 (Eq. 7 and lines 255-256), the correct number of negatives should be $M$ instead of $M-1$. We have corrected this typo in the revised manuscript. Also, given the constraint $j\neq i$ in Eq. 7, the superscript of the summation should indeed be $M+1$. In Algorithm 1 on p6, we have also corrected a typo: Each mini-batch should consists of $N$ positives and $N\cdot M$ negatives. This is because we use a memory bank, storing M negatives during training ($M$ is larger than the mini-batch size $N$). Therefore, this results in a total of $N \cdot M$ negatives per batch. It is important to note that there is no dependency between $N$ and $M$. We have further revised Algorithm 1 and unified the terminology in Sec 3.2 and Sec 3.3 to standardize all notation (marked red in the manuscript). Thank you for your careful review and constructive feedback.
>
>
> >(6) **W6:** *p5 254 what is the superscript of j in Eq. (8)? Is it the same as Eq. (7) or Eq. (3)?*
>
> Eq. 8 represents the binary Bayesian expression used within the softmax function to compute the conditional probability of a class given an observation $s$. Thus, the superscript of $j$ is 2, and the denominator represents the total probability of being either a clone negative or not.
>
> In the revised manuscript, we have further refined Eq. 8 to $$p\left(\mathcal{C} \mid s\right)=\frac{p\left(s\mid \mathcal{C}\right) p\left(\mathcal{C}\right)}{p\left(s \mid \mathcal{C}\right) p\left(\mathcal{C}\right) + p\left(s \mid \bar{\mathcal{C}}\right) p\left(\bar{\mathcal{C}}\right)}=\frac{\exp \left(a_{c}\right)}{\exp \left(a_{c}\right) + \exp \left(a_{\bar{c}}\right)}.$$
> Specifically, we change $p\left(\mathcal{C}=i \mid s\right), i \in \{0,1\}$ to $p\left(\mathcal{C}\mid s\right)$ and $p\left(\bar{\mathcal{C}}\mid s\right)$, which is much more beautiful and concise. $\mathcal{C}$ represents the pairwise similarity is a clone negative, and $\bar{\mathcal{C}}$ for not being a clone negative. We believe this revision eliminates potential misunderstandings arising from overlapping variable names.

---

> ### Author Response · Authors · 2024-11-20
> **Official reply to Reviewer PFrK [2/2]**
>
> >(7) **W7:** *p5 263 Why Eq. (9) is that? Can you give us a concrete deduction? And what about p(C=0|s)? Can you give a specific formulation?*
>
> We would be happy to discuss the derivation of Eq. 9 in more detail. To begin with, the $K$-class classification probability with Bayes's formula can be expressed as:
>
> $$p(y=i \mid x)=\frac{p(x \mid y=i) p(y=i)}{\sum_{j=1}^K p(x \mid y=j) p(y=j)}=\frac{\exp \left(f_i(x)\right)}{\sum_{j=1}^K \exp \left(f_j(x)\right)},$$
>
> In anchor selection of AdaCL, the input variable $x$ (a.k.a $s$) is one-dimensional with a binary output variable $y \in {0, 1}$ (a.k.a $\bar{\mathcal{C}}$ and $\mathcal{C}$). We aim to predict $p(y=1 \mid x)$. Gaussian Discriminant Analysis (GDA) assumes that for each class $y =0$ and $y=1$, the input $x$ follows a Gaussian distribution. This can be expressed as: $p(x \mid y=0)=\mathcal{N}\left(x \mid \mu_0, \sigma_0^2\right)$ and $p(x \mid y=1)=\mathcal{N}\left(x \mid \mu_1, \sigma_1^2\right)$. $\mu_0$, $\mu_1$ and $\sigma_0^2$, $\sigma_1^2$ are the means and variances of distributions for classes $y=0$ and $y=1$, respectively. Thus, the posterior probability can be expressed as:
> $$p(y=0 \mid x)=\frac{\mathcal{N}\left(x \mid \mu_0, \sigma_0^2\right) \cdot \pi_0}{\mathcal{N}\left(x \mid \mu_0, \sigma_0^2\right) \cdot \pi_0+\mathcal{N}\left(x \mid \mu_1, \sigma_1^2\right) \cdot \pi_1}, $$
>
>
> $$p(y=1 \mid x)=\frac{\mathcal{N}\left(x \mid \mu_1, \sigma_1^2\right) \cdot \pi_1}{\mathcal{N}\left(x \mid \mu_1, \sigma_1^2\right) \cdot \pi_1+\mathcal{N}\left(x \mid \mu_0, \sigma_0^2\right) \cdot \pi_0}. $$
>
> Since $\mathcal{N}\left(x \mid \mu_1, \sigma_1^2\right)$ is the probability density function of a gaussian distribution, we substitute $\mathcal{N}\left(x \mid \mu_1, \sigma_1^2\right) = \frac{1}{\sqrt{2 \pi \sigma_1^2}} \exp \left( -\frac{(x - \mu_1)^2}{2 \sigma_1^2} \right)$ into the above equations, obtaining Eq. 9 in the manuscript:
>
> $$p\left(y=0 \mid x\right)=\frac{1}{1+\frac{\pi_1}{\pi_0}\frac{\sigma_0}{\sigma_1}\exp\left[ \frac{\left( s - \mu_0\right)^2}{2\sigma_{0}^2} -\frac{\left( s - \mu_1\right)^2}{2\sigma_{1}^2}\right]},$$
>
> $$p\left(y=1 \mid x\right)=\frac{1}{1+\frac{\pi_0}{\pi_1}\frac{\sigma_1}{\sigma_0}\exp\left[ \frac{\left( s - \mu_1\right)^2}{2\sigma_{1}^2} -\frac{\left( s - \mu_0\right)^2}{2\sigma_{0}^2}\right]}.$$
>
> The equations represent the probability of a similarity score to be a clone negative or not, without the need of explicit pre-processing to the dataset or training. In the revised manuscript, we have added the derivation to the appendix. We hope the above analysis eliminate the ambiguity and strengthen the clarity of our paper.
>
>
>
> >(8) **W8:** *p10 509 How to correctly understand the heat map, since a sentence contain ALL WORDS, and every mentioned instance in the image SHOULD be highlighted. So, what does these Attention Map mean indeed? What Information can I get from them? Or what should we expect from them?*
>
>
> The purpose of the attention map is to reveal the specific regions in the image that align semantically with key parts of the sentence. Apart from the effectiveness of highlighting instances, Fig. 5 also illustrates the effectiveness of AdaCL's representation learning capacity when handling three clone negatives under different learning objectives. We assess the representation learning capacity by visualizing the attention map at early training stage (epoch 10). Our results show that AdaCL demonstrates superior and faster learning capacity compared with CL and TRL. For example, the background skyscrapers in case (1), and the man speaking to a blow horn in case (3) are quite significant in distinguishing clone negatives. This visualization provides qualitative insights into AdaCL's effectiveness.

---

### Official Review · Reviewer_5tK1 · 2024-11-03

**Soundness:** 2
**Presentation:** 2
**Contribution:** 2
**Rating:** 3
**Confidence:** 5

**Summary:**

This paper addresses a challenge in image-text matching referred to as clone negatives, which are negative image-text pairs that are semantically consistent with the positive pairs, leading to ambiguous and suboptimal matching results. To tackle this issue, the authors propose Adaptive Contrastive Learning (AdaCL), which employs two margin parameters with a modulating anchor to dynamically enhance the compactness between positives while mitigating the influence of clone negatives. The modulating anchor is selected based on the distribution of negative samples without explicit training, facilitating progressive tuning and improved in-batch supervision. Experiments on two benchmark datasets underscore the effectiveness of AdaCL.

**Strengths:**

1. The paper introduces a novel approach to address the issue of clone negatives in image-text matching, which is a pertinent problem in the domain.
2. The proposed Adaptive Contrastive Learning (AdaCL) framework dynamically adjusts the compactness between positives, showing potential for enhanced performance in image-text matching tasks.

**Weaknesses:**

1. The experimental results are not convincing. The main contribution of the paper is the improvement of Contrastive Learning, with CLIP being the most representative of this field. To prove the effectiveness of the paper, a comparison within the OpenCLIP framework using larger datasets (e.g., LAION) should have been conducted, rather than experiments on the relatively small-scale COCO dataset.
2. The proposed method appears to be somewhat complex, which goes against the inherently simple and scalable nature of contrastive learning. Additionally, the paper does not provide an analysis of the introduced hyperparameters or the potential increase in overall complexity.
3. The authors do not clearly explain why clone negatives pose a problem within the paradigm of contrastive learning. Clone negatives refer to texts that are aligned with the image but are not as accurate as the ground-truth text. Hence, the logits of clone negatives should naturally be lower than those of the ground-truth, which aligns with the basic optimization framework of contrastive learning. The authors should provide more empirical evidence or theoretical analysis demonstrating why standard contrastive learning fails to handle clone negatives adequately.

**Questions:**

The method figure (Figure 2) is unclear, and neither the figure caption nor the text adequately explains what the "reference clone negatives" refer to.

---

> ### Author Response · Authors · 2024-11-20
> **Official reply to Reviewer 5tK1 [1/2]**
>
> Thank you very much for the critical review of our work and your help in improving it. Here we provide a point-by-point response.
>
>
> >(1) **W1:** *The experimental results are not convincing. The main contribution of the paper is the improvement of Contrastive Learning, with CLIP being the most representative of this field. To prove the effectiveness of the paper, a comparison within the OpenCLIP framework using larger datasets (e.g., LAION) should have been conducted, rather than experiments on the relatively small-scale COCO dataset.*
>
> We sincerely appreciate your constructive suggestion. In response, we have conducted additional experiments as recommended. For further details, please refer to our Meta Reply.
>
> >(2) **W2:** *The proposed method appears to be somewhat complex, which goes against the inherently simple and scalable nature of contrastive learning. Additionally, the paper does not provide an analysis of the introduced hyperparameters or the potential increase in overall complexity.*
>
>
> We acknowledge that AdaCL introduces additional components, thus we have carefully analyzed its convergence efficiency in Appendix A.8 of the original manuscript. The experiments demonstrate that AdaCL maintains high convergence efficiency, ensuring its practicality in real-world applications. Furthermore, AdaCL can be seamlessly integrated into existing pipelines (image-text matching and pre-training) with minimal architectural modifications, making it a scalable and practical solution for a wide range of tasks.
>
> Regarding the computational overhead in AdaCL, here we further compare AdaCL with vanilla CL. Assume we have a mini-batch of $N$ samples, and number of negatives for each sample is $M$, the computation comparison can be expressed as follows:
>
> |        Operation        | Vanilla CL | AdaCL | Overhead |
> |:-----------------------:|:----------:| :-----: | :--------: |
> | Similarity Coomputation |  $N\cdot M$            | $N\cdot M$       |  0        |
> |    Anchor Selection                     | N/A           | $O(N\cdot M)$      |  $+O(N\cdot M)$        |
> | Margin Tuning ($m_1$, $m_2$)       |    N/A     |Scalar updates per batch       | Negligible     |
>
> We can conclude that AdaCL introduces moderate computational overhead, primarily due to anchor selection. The additional cost scales linearly with $N$ and $M$. We also record the training time of AdaCL in CLIP pre-training, which is demonstrated as follows:
>
> |     Learning objective     | # samples per second per GPU |
> |:---------------:|:--------------------:|
> |   CLIP+CL   |     462         |
> |  CLIP+AdaCL  |     411            |
>
>
> We record the average number of samples processed per second over the entire training process. We observe a slight decrease in efficiency for AdaCL compared to vanilla CL in the CLIP pre-training, which we recognize this as a limitation of AdaCL and have added this analysis to A.12 of the Appendix. Also, we believe this trade-off is acceptable given the context of pre-training, and AdaCL maintains a overall balance between effectiveness and efficiency. The above results and analysis have been updated to our revised manuscript to provide a thorough analysis of this work.

---

> ### Author Response · Authors · 2024-11-20
> **Official reply to Reviewer 5tK1 [2/2]**
>
> >(3) **W3:** *The authors do not clearly explain why clone negatives pose a problem within the paradigm of contrastive learning. Clone negatives refer to texts that are aligned with the image but are not as accurate as the ground-truth text. Hence, the logits of clone negatives should naturally be lower than those of the ground-truth, which aligns with the basic optimization framework of contrastive learning. The authors should provide more empirical evidence or theoretical analysis demonstrating why standard contrastive learning fails to handle clone negatives adequately.*
>
>
> The key limitation of standard contrastive learning lies in its **uniform treatment of all negative samples**, regardless of their similarity to the anchor. Clone negatives, as defined, are texts/images that are aligned with the image/text but are less accurate than the ground-truth. Due to their inherent similarity to the ground-truth text, clone negatives can exert a disproportionately strong influence during optimization, potentially leading to suboptimal representation learning.
>
>
> To address this limitation, our proposed AdaCL introduces an adaptive contrastive learning framework, where negatives with higher similarity to the anchor (e.g., clone negatives) are dynamically emphasized. By modulating the margin, AdaCL is able to force the model to pull them **further apart** from the anchor. AdaCL better encourages the model to learn finer-grained distinctions between semantically close representations.
>
> In Fig. 4 of the manuscript, we included a case study to compare AdaCL with standard contrastive learning, demonstrating its effectiveness. We acknowledge that a theoretical analysis, as the reviewer rightly points out, could further strengthen our argument. We appreciate this valuable suggestion and plan to explore it in the future work.
>
> >(4) **Q1:** *The method figure (Figure 2) is unclear, and neither the figure caption nor the text adequately explains what the "reference clone negatives" refer to.*
>
> In the revised manuscript, we have further included a more detailed explanation of Fig. 2 to address the your concerns. Specifically, reference clone negatives are those selected based on the Salient Score introduced in Sec. 3.3. Similarly, salient negatives are also identified through sorting by the Salient Score. We hope this clarification and the revised manuscript provides a clearer understanding of the concepts depicted in Fig. 2.

---

> ### Author Response · Authors · 2024-11-24
>
> Dear reviewer 5tK1,
>
> Thank you so much for your time and efforts in reviewing our paper. We have addressed your comments in detail and are happy to discuss more if there are any additional concerns. We are looking forward to your feedback and would greatly appreciate you consider raising the scores.
>
> Best regards,
>
> Authors

---

> > ### Comment · Reviewer_5tK1 · 2024-11-25
> >
> > Thanks for your reply and additional experiments.
> > What concerns me is the computational overhead. When training CLIP on large-scale datasets using GPU clusters, the added computational burden becomes increasingly significant. In such scenarios, employing an efficient algorithm capable of leveraging larger batch sizes tends to be more practical and impactful.
> > Consequently, many algorithms proposed to enhance contrastive learning face challenges in real-world CLIP training due to their inefficiency. This limitation is also reflected in your experiments, where the improvement observed on CC12M is notably less substantial compared to CC3M.

---

> > > ### Author Response · Authors · 2024-11-25
> > > **Official reply to Reviewer 5tK1**
> > >
> > > Thank you for your valuable feedback. Below we address your concerns point-by-point. Regarding computational overhead, beyond CLIP pretraining, AdaCL's plug-and-play effectiveness has been demonstrated across various downstream tasks on multiple baselines, which are minimally affected by computational overhead. This makes the performance improvements brought by AdaCL **a highly acceptable trade-off**. As for batch size, integrating AdaCL does not increase GPU utilization, ensuring that **the batch size remains unaffected by its implementation**.
> > >
> > > Also, we would like to take this opportunity to clarify that our primary research focus is to **elucidate the concept of clone negatives and to effectively address this issue within the scope of image-text matching**. While we have validated AdaCL on large-scale datasets, pre-training is not the key aspect of this work. Nonetheless, we fully recognize its importance and intend to explore real-world CLIP training further in future research.
> > >
> > > We hope this clarification helps highlight the contributions of our paper and its potential to advance research in image-text matching.

---

### Author Response · Authors · 2024-11-20
**Meta Reply [2/2]**

In addition to image-text matching, we also evaluate AdaCL on more downstream tasks, i.e., text-based person search and zero-shot classification, where the former has been detailed in the manuscript. For zero-shot classification, we have verified AdaCL on eight common classification benchmarks, which can be divided into (i) general datasets: ImageNet[2], CIFAR-10[3], CIFAR-100[3], Caltech-101[4]), and (ii) fine-grained datasets: Food-101[5], Flowers-102[6], OxfordPets[7], and FGVCAircraft[8]. The Top-1 accuracy results of "CLIP + AdaCL" pretrained on CC3M and CC12M are as follows:

|  Pretrain Data  |      Method      |     ImageNet     |      CIFAR-10       |       CIFAR-100       |     Caltech-101      |      Food-101       |  Flowers  |    Pets    | Aircraft    |
|:---------------:|:----------------:|:----------------:|:-------------------:|:---------------------:|:--------------------:|:-------------------:|:---------:|:----------:| :---: |
|      CC3M       |     Baseline     |       17.2       |        71.3         |         32.1          |         50.9         |        10.2         |   10.8    |    12.1    | 1.0    |
|      CC3M       |      **AdaCL**       |       22.0       |        77.1         |         42.2          |         54.8         |        12.6         |   13.3    |    14.9    | 1.7    |
| --------------- | ---------------- | ---------------- | ------------------- | --------------------- | -------------------- | ------------------- | --------- | ---------- | ---------    |
|      CC12M      |     Baseline     |       32.9       |        72.5         |         38.0          |         74.0         |        26.5         |   25.7    |    46.2    | 2.6    |
|      CC12M      |      **AdaCL**       |       34.8       |        73.4         |         43.3          |         74.7         |        33.1         |   25.4    |    46.7    | 2.8    |

AdaCL outperforms CL in all the general datasets and most of the fine-grained datasets, proving its advantage in recognition tasks. Noted that the performance on fine-grained datasets further verifies AdaCL's capacity in challenging scenarios. We want to emphasize that the simple-yet-effective design of AdaCL (without the need of explicit modules or training) verifies its potential in image-text contrastive learning. All the additional experiments have been added to the revised manuscript to strengthen its overall quality.


So far, AdaCL has been validated across (1) multiple image-text matching baselines, (2) CLIP pre-training, (3) weakly-supervised image-text matching. Additionally, AdaCL has demonstrated strong performance on other downstream tasks, including (4) text-based person search and (5) zero-shot image classification. It is noteworthy that our proposed weakly-supervised matching scenario showcases its great potential in training without crowd-sourced annotations, which is another contribution of this paper. As one of the fundamental vision-language tasks, image-text matching serves as the most suitable benchmark for assessing and addessing clone negatives, which is a critical limitation that vanilla contrastive learning frameworks struggle to handle. Through extensive experiments across diverse tasks, we believe the effectiveness, compatibility, scalability, and robustness of AdaCL have been comprehensively validated.

[1] Sharma et al., Conceptual captions: A cleaned, hypernymed, image alt-text dataset for automatic image captioning. In: Association for Computational Linguistics (2018)

[2] Deng et al., Imagenet: A large-scale hierarchical image database. In: IEEE Conf. Comput. Vis. Pattern Recog. (2009)

[3] Krizhevsky et al., Learning multiple layers of features from tiny images (2009)

[4] Fei-Fei et al., Learning generative visual models from few training examples: An incremental bayesian approach tested on 101 object categories. In: IEEE Conf. Comput. Vis. Pattern Recog. (2004)

[5] Bossard et al., Food-101–mining discriminative components with random forests. In: Eur. Conf. Comput. Vis. (2014)

[6] Nilsback et al., Automated flower classification over a large number of classes. In: Sixth Indian Conference on Computer Vision, Graphics & Image Processing (2008)

[7] Parkhi et al., Cats and dogs. In: Int. Conf. Comput. Vis. (2012)

[8] Maji et al., Fine-grained visual classification of aircraft. arXiv:1306.5151 (2013)

---

### Author Response · Authors · 2024-11-20
**Meta Reply [1/2]**

We would like to thank all the reviewers for their thorough reviews and highly valuable feedback and suggestions. We are excited to hear that the reviewers found our work pertinent and meaningful [R1, R2, R3, R4], well-motivated and novel [R2, R3], and experimentally comprehensive [R2, R3, R4].


We also appreciate the suggestions provided, which have strengthened our work. Following their suggestions, we improved our paper accordingly (see latest revised manuscript). Below, we provide an answer to the common point raised about pre-training.

**Pre-training and comparison with CLIP**

In order to validate the pre-training ability of AdaCL, we conduct experiments in contrastive image-text pre-training under two large-scale dataets. Due to the time limit, we report the pre-training results on Conceptual Captions 3M (CC3M), and CC12M datasets [1]. Compared to MSCOCO and Flickr30K, they represent the scalability in larger-scale settings. Specifically, We use ViT-B/32 as image backbone, and set "CLIP + vanilla CL" as the baseline. "AdaCL" denotes pre-training with "CLIP + AdaCL". The following table is the zero-shot image-text matching results on the testset of Flickr30k dataset:


|  Pretrain Data  |      Method      |      I-T R@1       |      I-T R@5       |       I-T R@10       |      T-I R@1       |      T-I R@5       |       T-I R@10       |
|:---------------:|:----------------:|:-------------------:|:-------------------:|:---------------------:|:-------------------:|:-------------------:|:---------------------:|
|      CC3M       |     Baseline     |        26.6         |        52.5         |         63.2          |        18.1         |        39.4         |         49.7          |
|      CC3M       |      AdaCL       |  39.5                   | 60.8                    |  73.7                     |   25.5                  |  46.9                   |   54.3                    |
| --------------- | ---------------- | ------------------- | ------------------- | --------------------- | ------------------- | ------------------- | --------------------- |
|      CC12M      |     Baseline     |        49.3         |        77.3         |         85.0          |        35.5         |        61.8         |         71.6          |
|      CC12M      |      AdaCL       | 51.0                    | 77.5                    | 87.9                      |  38.4                   | 64.6                    |  74.7                     |

The second table is the zero-shot image-text matching results on the testset of MSCOCO dataset:


|  Pretrain Data  |      Method      |      I-T R@1       |      I-T R@5       |       I-T R@10       |      T-I R@1       |      T-I R@5       |       T-I R@10       |
|:---------------:|:----------------:|:-------------------:|:-------------------:|:---------------------:|:-------------------:|:-------------------:|:---------------------:|
|      CC3M       |     Baseline     |        13.4         |        32.0         |         43.3          |        10.1         |        25.6         |         35.7          |
|      CC3M       |      AdaCL       | 22.5                    | 47.1                    | 60.7                      | 17.8                    | 31.6                    |  39.5                     |
| --------------- | ---------------- | ------------------- | ------------------- | --------------------- | ------------------- | ------------------- | --------------------- |
|      CC12M      |     Baseline     |        29.3         |        54.4         |         65.3          |        19.0         |        41.0         |         52.5          |
|      CC12M      |      AdaCL       |  34.0                   | 55.6                    | 65.9                      |  25.1                   |  47.3                   |  57.4                     |

"I-T" and "T-I" represent image-text matching and text-image matching, respectively. From the results above, it can be observed that AdaCL improves CLIP's performance compared to vanilla CL, leading to notable enhancements in both image-text and text-image matching.

---

### Meta-Review · Area_Chair_Pxxu · 2024-12-18

**Metareview:**

This paper focuses on the clone negatives problem in image-text matching and proposes Adaptive Contrastive Learning (AdaCL), which introduces two margin parameters to dynamically strengthen the compactness between positives. Extensive experiments image-text matching, noisy correspondence learning, CLIP pre-training, and text-based person search, are conducted to prove the effectiveness of AdaCL. The introduced approach might incur extra costs for training. However, given the strong results from the paper, the increase of cost (presented by # samples per second per GPU) is negligible.

**Additional Comments On Reviewer Discussion:**

Reviewers PFrK and 8azX raised questions about the writing of the paper, which were addressed in detail by the authors. Reviewer vmnW asked for more recent CLIP models and the authors added CLIP + AdaCL to the study. Reviewer 5tK1 asked for the increased cost and larger-scale experiments. These were all answered by the authors.

---

### Decision · Program_Chairs · 2025-01-22

Accept (Poster)